



**Understanding influence of ocean waves on Arctic sea ice simulation: A modeling study**
**with an atmosphere-ocean-wave-sea ice coupled model**
**Chao-Yuan Yang[1], Jiping Liu[2], Dake Chen[1]**
[1]Southern Marine Science and Engineering Guangdong Laboratory (Zhuhai), Zhuhai,
Guangdong, China
[2]Department of Atmospheric and Environmental Sciences, University at Albany, State
University of New York, Albany, NY, USA
Corresponding authors:
Chao-Yuan Yang (yangchaoyuan@sml-zhuhai.cn) and Jiping Liu (jliu26@albany.edu)





**Abstract**
Rapid decline of Arctic sea ice has created more open water for ocean wave development
and highlighted the importance of wave-ice interactions in the Arctic. Some studies have made
contributions to our understanding of the potential role of the prognostic floe size distribution
(FSD) on sea ice changes. However, these efforts do not represent the full interactions across
atmosphere, ocean, wave, and sea-ice. In this study, we implement a modified joint floe size
and thickness distribution (FSTD) in a newly-developed regional atmosphere-ocean-wave-sea
ice coupled model and conduct a series of pan-Arctic simulation with different physical
configurations related to FSD changes, including FSD-fixed, FSD-varied, lateral melting rate,
wave-fracturing formulation, and wave attenuation rate. Firstly, our atmosphere-ocean-wave-
sea ice coupled simulations show that the prognostic FSD leads to reduced ice area due to
enhanced ice-ocean heat fluxes, but the feedbacks from the atmosphere and the ocean partially
offset the reduced ice area induced by the prognostic FSD. Secondly, lateral melting rate
formulations do not change the simulated FSD significantly but they influence the flux
exchanges across atmosphere, ocean, and sea-ice and thus sea ice responses. Thirdly, the
changes of FSD are sensitive to the simulated wave height, wavelength, and wave period
associated with different wave-fracturing formulations and wave attenuation rates, and the
limited oceanic energy imposes a strong constraint for the response of sea ice to FSD changes.
Finally, our results also demonstrate that wave-related physical processes can have impacts on
sea ice changes with the constant FSD, suggesting the indirect influences of ocean waves on
sea-ice through the atmosphere and the ocean.



## 1. Introduction

Arctic sea ice, a major component in the climate system, has undergone dramatic changes over the past few decades associated with global climate change. September and March Arctic sea ice extent shows decreasing trends of -13.1% and -2.6% per decade from 1979 to 2020, respectively (Perovich et al., 2020). The mean Arctic sea ice thickness has decreased by ~1.5-2 meters from the submarine period (1958-1976) to the satellite period (2011-2018), largely resulting from the loss of multiyear ice (Kwok, 2018; Tschudi et al., 2016). The drifting speed of Arctic sea ice exhibits an increasing trend based on satellite and buoy observations (e.g., Rampal et al., 2009; Spreen et al., 2011; Zhang et al., 2022). As the Arctic Ocean has been dominating by thinner and younger ice, Arctic sea ice is more likely to be influenced by forcings from the atmosphere and the ocean.

Associated with the above Arctic sea ice changes, the Arctic fetch (open water area for ocean wave development) is less limited by the ice cover. The increased Arctic fetch and surface wind speed are able to lead to higher ocean waves in the Arctic Ocean based on observations, reanalysis, and future projections (Casas-Prat and Wang, 2020; Dobrynin et al., 2012; Liu et al., 2016; Stopa et al., 2016; Waseda et al., 2018). The higher ocean waves are more likely to propagate deeper into the ice pack and have sufficient energy to break sea ice into smaller floes (e.g., Kohout et al., 2014). Sea ice with mostly smaller floes has larger surface area, particularly lateral surface. The increased lateral surface accelerates ice melting through enhanced ice-ocean heat fluxes (e.g., Steele, 1992). Some studies also showed that the ice-floe melting rate is a result of the interaction between floe size and ocean circulation (Gupta and



Thompson, 2022; Horvat et al., 2016). The enhanced ice melting creates more open water (i.e.,
fetch), which is a favorable condition for further wave development as well as the ice-albedo
feedback (Curry et al., 1995). These processes create a potential feedback loop between ocean
waves and sea ice (e.g., Asplin et al., 2014; Thomson and Rogers, 2014).

Arctic cyclones and their high surface wind are the important driver for large wave events

in the Arctic Ocean. Previous studies showed that intense storms like "Great Arctic Cyclone"
of 2012 (Simmonds and Rudeva, 2012) and strong summer cyclone in 2016 contribute to the
anomalously low sea ice extent in 2012 and 2016 (e.g., Lukovich et al., 2021; Parkinson and
Comiso, 2013; Peng et al., 2021; Stern et al., 2020; Zhang et al., 2013). Statistical analyses
based on cyclone-tracking algorithm across multiple reanalyses suggested that the number of
Arctic cyclones show a significantly positive trend in the cold season (e.g., Sepp and Jaagus,
2011; Valkonen et al., 2021; Zahn et al., 2018). The increased cyclone activities and more open
water areas cause more extreme wave events in the Arctic (e.g., Waseda et al., 2021).
Blanchard-Wrigglesworth et al. (2021) found that extreme changes in Arctic sea ice extent are
correlated with distinct wave conditions during the cold season based on the observations.

The potential feedback loop associated with ocean waves and sea ice and more extreme

wave events indicate the importance to represent these processes in climate models for
improving sea ice simulation and prediction (e.g., Collins et al., 2015; Kohout et al., 2014).
However, state-of-the-art climate models participating in the latest Coupled Model
Intercomparison Project Phase 6 (CMIP6) have not incorporated the interactions between
ocean waves and sea ice in their model physics (e.g., Horvat, 2021). The coupled effects of



ocean waves and sea ice include; the amplitude of ocean waves decays as the waves travel
under the ice cover due to the combination of scattering and dissipation (e.g., Squire, 2020).
Crests and troughs of ocean waves exert strains to sea ice, and sea ice breaks if the maximum
strain exceeds certain threshold (e.g., Dumont et al., 2011). The wave-induced ice breaking
changes the size of floes, which in turn changes the floe size distribution (FSD; Rothrock and
Thorndike, 1984). In addition to the interactions between ocean waves and sea ice, the floe size
contributes to the changes of atmospheric boundary layer (e.g., Schafer et al., 2015; Wenta and
Herman, 2019), mechanical responses of sea ice (e.g., Vella and Wettaufer, 2008; Weiss and
Dansereau, 2017; Wilchinsky et al., 2010), the flux exchanges across air-sea ice-ocean
interfaces (Cole et al., 2017; Loose et al., 2014; Lu et al., 2011; Martin et al., 2016; Steele et
al., 1989; Tsamados et al., 2014), and the scattering of ocean wave propagation (e.g., Montiel
et al., 2016; Squire and Montiel, 2016). Thus, it is essential to have a prognostic FSD to
properly reflect wave-ice interactions as well as other processes related to the floe size in
climate models.

Recently, several studies have made contributions on understating responses of sea ice to

the prognostic FSD (e.g., Bateson et al., 2020; Bennetts et al., 2017; Boutin et al., 2020; Horvat
and Tziperman, 2015; Roach et al., 2018a, 2019; Zhang et al., 2015, 2016). However, these
studies used simplified model complexity (i.e., standalone sea ice model, ice-wave coupling,
ice-ocean coupling) and unable to give a full representation of sea ice responses under the
interactions across atmosphere, ocean, wave, and sea ice. Motivated by this, here we introduce
a newly-developed atmosphere-ocean-wave-sea ice coupled model, in which we implement



physical processes that simulate the evolution of floe size distribution. We use this new coupled
model to investigate the responses of sea ice to ocean waves, as well as interactions in the
Arctic climate system. This paper is structured as follows. Section 2 provides an overview of
the new coupled model, focusing on the wave component and the implementation of the
prognostic FSD. Section 3 describes the design of numerical experiments and the related model
configurations. Section 4 examines the responses of sea ice to wave-ice interactions with the
prognostic FSD, as well as other ocean waves-related processes. Discussions and concluding
remarks are provided in section 5.

**2.    Model description**

The newly-developed atmosphere-ocean-wave-sea ice coupled model is based on

Coupled Arctic Prediction System (CAPS, Yang et al., 2022), which consists of the Weather
Research and Forecasting Model (WRF), the Regional Ocean Modeling System (ROMS), and
the Community Ice CodE (CICE). The detailed description of each model component in CAPS
is referred to Yang et al. (2020; 2022). In this section, we focus on newly-added features in
CAPS as described below.
**2.1.  Wave model component**

To represent wave-ice interactions, an ocean wave model is coupled into CAPS, which is

the Simulating Waves Nearshore (SWAN). SWAN is a third-generation wave model and
includes processes of diffraction, refraction, wave-wave interactions, and wave dissipation due
to wave breaking, whitecapping, and bottom friction (Booij et al., 1999). Recently, the SWAN





model has implemented wave dissipation due to sea ice based on an empirical formula, which
is called IC4M2 (Collins and Rogers, 2017; Rogers, 2019). Specifically, the temporal
exponential decay rate of wave energy due to sea ice is defined as,
$$S_{ice}/E = -2c_g k_i \ (1)$$

where $S_{ice}$ is the sink term induced by sea ice, $E$ is the wave energy spectrum, and $c_g$ is the

group velocity. $k_i$ is the linear exponential rate that is a function of frequency as follow,

$$k_i(f) = c_0 + c_1 f + c_2 f^2 + c_3 f^3 + c_4 f^4 + c_5 f^5 + c_6 f^6 \ (2)$$

where $c_0$ to $c_6$ are the user-defined coefficients and their values as described in Section 3. In
the SWAN model, both the wind source term $S_{in}$, and the sea ice sink term are scaled by sea
ice concentration $a_{ice}$, which is provided by the CICE model through the coupler in CAPS,
$$S_{ice} \rightarrow a_{ice} S_{ice} \ (3)$$

$$S_{in} \rightarrow (1 - a_{ice}) S_{in} \ (4)$$

**2.2. Prognostic FSD**

For the prognostic FSD implemented in the CICE model, we follow the joint floe size and

thickness distribution (FSTD; Horvat and Tziperman, 2015). The FSTD is defined as a
probability distribution $f(r, h)drdh$. $f(r, h)$ represents the fraction of cell covered by ice with
floe size between $r$ and $r + \Delta r$, thickness between $h$ and $h + \Delta h$, and the FSTD satisfies,
$$\int_{\mathcal{R}} \int_{\mathcal{H}} f(r, h)drdh = 1 \ (5)$$

The ice thickness distribution $g(h)$ (ITD; Thorndike et al., 1975), which is simulated by the

CICE model, and the FSD $F(r)$, can be obtained by integrating the FSTD over all floe sizes



and all ice thicknesses,

$$\int_{\mathcal{R}} f(r,h)dr = g(h)$$
$$\int_{\mathcal{H}} f(r,h)dh = F(r) \tag{6}$$

143

Roach et al. (2018a) suggested the modified FSTD, $L(r,h)$, to preserve the governing

equations of ITD in the CICE model, which satisfies,

$$\int_{\mathcal{R}} L(r,h)dr = 1 \tag{7}$$

149

and

$$f(r,h) = g(h)\,L(r,h) \tag{8}$$

As described in Roach et al. (2018a), the implementation of the modified FSTD ignores the
two-way relationship between floe size, that is, physical processes associated with FSD
changes (i.e., $L(r,h)$ changes) are independent across each ice thickness category. The
governing equation of FSTD is defined as,

$$\frac{\partial f(r,h)}{\partial t} = -\nabla \cdot (f(r,h)\vec{v}) + \mathcal{L}_T + \mathcal{L}_M + \mathcal{L}_W \tag{9}$$

The terms in the right-hand-side represent advection, thermodynamics, mechanical, and wave-
induced floe-fracturing processes. For these terms, except the last term $\mathcal{L}_W$, we follow the
approach described in Roach et al. (2018a) and related values for coefficients as described in
Section 3. The formulations of $\mathcal{L}_W$ proposed in Horvat and Tziperman (2015) involves a
random function to generate sub-grid scale sea surface elevation to determine how floes are
fractured by ocean waves. As a consequence, simulations are not bitwise reproducible with the
formulation including a random function. To avoid this issue, we propose different approaches





for our implementation of FSTD as described below.
**2.3. Floe fracturing by ocean waves**
For the floe-fracturing term $\mathcal{L}_W$, we follow the formulation suggested by Zhang et al.
(2015), which has similar form as Horvat and Tziperman (2015) and can be described as,

$$168 \qquad \mathcal{L}_W = -Q(r)\,f(r,h) + \int_{\mathcal{R}} \beta(r',r)Q(r')f(r',h)dr' \quad (10)$$

The first term in the right-hand-side represents the areal fraction reduction due to floe-
fracturing and the second term is the areal fraction gain from other floe size categories that
have floe-fracturing. In equation (10), $Q(r)$ is the probability that floe-fracturing occurs for
floe size between $r$ and $r + \Delta r$, and $\beta(r',r)$ is the redistributor that transfers fractured floe
from floe size $r'$ to $r$. $\mathcal{L}_W$ does not create or destroy ice so it must satisfy,
$$\int_{\mathcal{R}} \mathcal{L}_W\, dr = 0 \quad (11)$$
In this study, we propose two different formulations for $Q(r)$ and $\beta(r',r)$.
(a) Equally-redistribution
We follow the same assumption in Zhang et al. (2015). That is, ice-fracturing by ocean
waves is likely to be a random process and the size of fractured floe does not have favored floe
size based on aerial photographs and satellite images (e.g., Steer et al., 2008; Toyota et al.,
2006, 2011). Thus, fractured floe is equally-redistributed into smaller floe sizes. The
redistributor is defined as,
$$\beta(r_1,r_2) = \begin{cases} 1/(c_2 r_1 - c_1 r_1) & if\ c_1 r_1 \leq r_2 \leq c_2 r_1 \\ 0 & if\ r_2 < c_1 r_1\ or\ r_2 > c_2 r_1 \end{cases} \quad (12)$$

where $c_1$ and $c_2$ are constants that define upper- and lower-bound of floe size redistribution.





Details of $\beta(r',r)$ in this formulation are referred to Zhang et al. (2015).

For the probability $Q(r)$, Zhang et al. (2015) used an user-defined coefficient to reflect

wave conditions and determine $Q(r)$. Zhang et al. (2016) suggested that the coefficient is a

function of wind speed, fetch, ITD, and FSD. Since CAPS has a wave component to simulate

wave conditions, we reformulate $Q(r)$ to include simulated wave information from the coupler

and $Q(r)$ is defined as,

$$Q(r) = c_w H(\varepsilon) exp\left[-\propto\left(\frac{1-r}{r_{max}}\right)\right] \quad (13)$$

where $H(\varepsilon)$ is the Heaviside step function, the exponential function determines the fraction of
each floe size participating in fracturing, and user-defined coefficients, $c_w$ and $\propto$, control the
upper-bound of $Q(r)$ and the shape of the exponential function. To include wave conditions
from the SWAN model, we apply the floe-fracturing parameterization suggested by Dumont et
al. (2011) to calculate the strain induced by ocean waves on ice floes, and use this
parameterization to define $H(\varepsilon)$ as,

$$H(\varepsilon) = \begin{cases} 1, & if\ \varepsilon \geq \varepsilon_c \\ 0, & if\ \varepsilon < \varepsilon_c \end{cases} \quad (14)$$

$$\varepsilon = \frac{2\pi^2 h_{ice} A_{wave}}{L_{wave}^2} \quad (15)$$

where the strain $\varepsilon$ is proportional to the ice thickness $h_{ice}$ and the mean amplitude of wave
$A_{wave}$, and inversely proportional to the square of the mean surface wavelength $L_{wave}$. If the
strain exceeds the strain yield limit $\varepsilon_c$ (see Section 3), floe-fracturing occurs (i.e., $H(\varepsilon) = 1$).
The distribution of wave heights is, in general, a Rayleigh distribution, which allows us to use
the simulated significant wave height from the SWAN model to determine the mean wave
amplitude with following relationship (e.g., Bai and Bai, 2014),



$$A_{wave} = \frac{H_{wave}}{2} \cong \frac{5}{16}H_s \quad (16)$$
where $H_{wave}$ is the mean wave height, and $H_s$ is the significant wave height.
The exponential function is built on that the wave-strain on ice floes is separated by the
wavelength (e.g., Dumont et al., 2011, their Fig. 4). Floe size smaller than the wavelength is
more likely to move along with ocean waves with little bending (e.g., Meylan and Squire, 1994).
That is, the exponential function preferentially has higher fraction for larger floes.
(b) Redistribution based on a semi-empirical wave spectrum
As discussed in Dumont et al. (2011, their Fig. 4), fractured floes have a maximum size
with half of the surface wavelength. Thus, the wave distribution of different wavelengths in
each grid cells allows us to predict floe sizes after fracturing. The sea surface elevation is a
result of the superimposition of waves with different periods, amplitudes, and directions in
space and time. Empirical wave spectra have been proposed to describe wave conditions with
a finite set of parameters. Based on wave observations from a wide variety of locations,
Bretschneider (1959) suggested the formulation of wave spectrum, which are used to formulate
the redistribution of fractured-floe as described below.
The Bretschneider wave spectrum is defined as,
$$S_B(T) = \frac{1.25H_s{}^2T^5}{8\pi T_p{}^4}exp\left[-1.25\left(\frac{T}{T_p}\right)^4\right] \quad (17)$$

where $T_p$ is the peak wave period, and the spectral wave amplitude is defined as (Dumont et

al., 2011),

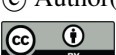


---

221
$$A(T) = \sqrt{\frac{4\pi S_B(T)}{T}} \quad (18)$$

Similar to the distribution of wave height, Bretschneider (1959) found that the distribution of

wave period is, in general, a Rayleigh distribution and defined as,

$$P(T) = 2.7\left(\frac{T}{T_{ave}}\right)^3 exp\left[-0.675\left(\frac{T}{T_{ave}}\right)^4\right] \quad (19)$$

where $T_{ave}$ is the mean surface period. With the deep-water surface wave dispersion relation
$L(T) = gT^2/2\pi$, the corresponding wave length for each wave period bin can be obtained,
and the wave-strain distribution can be calculated with the modified equation (15),
$$\varepsilon(T) = \frac{2\pi^2 h_{ice} A(T)}{L(T)^2} \quad (20)$$

Combined with the Heaviside step function defined in the equation (14), the probability of floe-

fracturing for each wave period is obtained,

$$P_f(T) = H\big(\varepsilon(T)\big)\overline{P}(T) \quad (21)$$

where $\overline{P}(T)$ is the normalized $P(T)$. Based on $P_f(T)$ and the assumption that fractured floes
have a maximum size with half of the surface wavelength, the redistributor $\beta(r_1, r_2)$ can be
obtained based on following criteria: 1) floe size between $r$ and $r + \Delta r$ (in radius) must be
greater than half of wavelength $L(T)$, 2) floes fractured by the wavelength $L(T)$ have the size
of $L(T)/2$, and 3) $P_f(T)$ represents the fraction of floe with $r$ and $r + \Delta r$ transferred to new
size with $r'$ and $r' + \Delta r$ determined by the criterion (2). The probability $Q(r)$ is the summation
of $P_f(T)$ and represents the total fraction of floe participating in wave-fracturing.
**3.    Model configurations and experiment designs**

The model domain includes 320 (440) x- (y-) grid points with a ~24km resolution for all





model components (Fig. 1). Initial and boundary conditions for the WRF, ROMS, CICE models
are generated from the Climate Forecast System version 2 (CFSv2, Saha et al., 2014)
operational analysis, archived by National Centers for Environmental Information (NCEI),
National Oceanic and Atmospheric Administration (NOAA). In our configurations, the SWAN
model starts with the calm wave states (i.e., zero wave energy in all frequencies). The modified
FSTD, $L(r, h)$, is initialized based on the power-law distribution of floe number, $N(r) \propto r^{-a}$
(e.g., Toyota et al., 2006), with the exponent $a$ as 2.1 for all grid cells. Physical
parameterizations of each model component are mostly identical to those used in Yang et al.
(2022) and summarized in Table 1.
Cassano et al. (2011) suggested that the use of a higher model top (10 mb) or applying
spectral nudging in the upper model levels lead to significantly reduced bases in pan-Arctic
atmospheric circulation in the standalone WRF model. Thus, compared with Yang et al. (2022),
we change the model top of the WRF model in CAPS from 50 mb to 10 mb. With coupling to
the SWAN model in CAPS, the corresponding configurations are modified to reflect wave
effects on the atmosphere and the ocean. In the Mellor-Yamada-Nakanishi-Niino planetary
boundary layer scheme (MYNN; Nakanishi and Nino, 2009), the surface roughness, $z_0$, is
modified to include the effect of waves based on the following formulation,
$$z_0 = 1200 H_s \left(\frac{H_s}{L_{wave}}\right)^{4.5} + \frac{0.11 \upsilon}{u_*} \quad (22)$$
where $\upsilon$ is the viscosity, and $u_*$ is the friction velocity (Taylor and Yelland, 2001; Warner et al.,
2010). For the interaction of ocean waves and currents, the vortex-force (VF) formulation is
applied that represents conservative (e.g., vortex and Stokes-Coriolis forces) and non-



conservative wave effects. The non-conservative wave effects in the VF formulation include
wave accelerations for currents and wave-enhanced vertical mixing (Kumar et al., 2012;
Uchiyama et al., 2010). The dissipated wave energy due to surface wave breaking and
whitecapping is transferred to the ocean surface layer as additional turbulent kinetic energy,
which in turn enhances the vertical mixing. For the effect of currents on the dispersion relation
in wave propagation, we employ a depth-weighted current to account for the vertically-sheared
flow following Kirby and Chen (1989). As discussed in previous studies (e.g., Naughten et al.,
2017; Yang et al., 2022), the upwind third-order advection (U3H, Table 1) scheme, which is an
oscillatory scheme, can lead to increased non-physical frazil ice formation. To address this
issue, we implement the upwind flux limiter suggested by Leonard and Mokhtari (1990) to
reduce false extrema caused by the oscillatory behavior of the U3H scheme. The value of
yielding strain $\varepsilon_c$, described in Section 2.3 is chosen as $\cong 3 \times 10^{-5}$ (Dumont et al., 2011;
Horvat and Tzipermann, 2015; Langhorne et al., 1998). The floe welding parameter in the
thermodynamic term $\mathcal{L}_T$, is chosen as $1 \times 10^{-7}\ km^{-2}s^{-1}$. Roach et al. (2018b) found a lower
bound of floe welding parameter as $1 \times 10^{-9}\ km^{-2}s^{-1}$ in the autumn Arctic based on the
observations. For the user-defined coefficients in the equation (4), all experiments use the
equally-redistributed formulation described in Section 2.3 with $c_w$ as 0.8 and $\propto$ as 1.0. Based
on the formation of $\mathcal{L}_T$ in the equation (9) (see Roach et al., 2018a), the floe size change
through the lateral surface is determined by both the floe size and the lateral melting rate. In
the existing sea ice models, the lateral melting rate $w_{lat}$ is all based on the empirical
formulation suggested by Perovich (1983, hereafter P83),





$$w_{lat} = m_1 \Delta T^{m_2} \quad (23)$$

where $\Delta T$ is the temperature difference between sea surface temperature (SST) and the freezing
point, and $m_1$, $m_2$ are empirical coefficients based on the observations from a single sea ice
lead in the Canadian Arctic. This empirical formulation is also the default lateral melting rate
in the CICE model. Maykut and Perovich (1987, hereafter MP87) showed a different approach
to parameterize the lateral melting rate that includes the friction velocity $u_*$ based on the
observations from the Marginal Ice Zone Experiment, which is defined as,

$$w_{lat} = u_* m_3 \Delta T^{m_4} \quad (24)$$

Both formulations (Equ. 23, 24) are examined in this study (see Table 2). In the equation (2),
the user-defined coefficients for the wave attenuation are set as $c_2 = 1.06 \times 10^{-3}$ and $c_4 =$
$2.3 \times 10^{-2}$ (case 1), which follow the polynomial of Meylan et al. (2014, hereafter M14) from
the observations with 10-25m floe in diameter in the Antarctic, and $c_2 = 2.84 \times 10^{-4}$ and
$c_4 = 1.53 \times 10^{-2}$ (case 2), which follow the polynomial of Rogers et al. (2018, hereafter R18)
based on the observations for pancake and frazil ice in the Arctic.
In this study, a series of numerical experiments for the pan-Arctic sea ice simulation have
been conducted, starting from January 1st 2016 to December 31st 2020. Table 2 provides the
details of the configurations for these experiments, which allow us to examine the influence of
ocean waves and related physical processes on Arctic sea ice simulation in the atmosphere-
ocean-wave-sea ice coupled framework. Specifically, these experiments focus on 1) the
comparison between constant FSD and prognostic FSD (Exp-CFSD and Exp-PFSD), 2) sea ice
responses to different lateral melting rate parameterizations (Exp-CFSD, Exp-PFSD, Exp-



LatMelt-C and Exp-LatMelt-P), 3) the difference between the equally-redistributed
formulation and the Bretschneider formulation for floe fracturing (Exp-PFSD and Exp-
WaveFrac-P), and 4) the contribution of different wave attenuation rates to sea ice changes
(Exp-CFSD, Exp-PFSD, Exp-WaveAtt-C and Exp-WaveAtt-P).
**4.  Results**
**4.1  Constant vs. Prognostic floe size**
Figure 2 shows the evolution of sea ice area (SIA) for all experiments conducted in this
study (as well as the values of seasonal maximum and minimum SIA for all experiments are
summarized in Table S1).  SIA is calculated as the sum of ice-covered area of all grid cells
(cell-area times sea ice concentration). In addition to the evolution of SIA, the 2016-2020
averaged March and September sea ice concentration (SIC) for all experiments are shown in
Figure S1. Compared with Exp-CFSD, which uses a constant floe diameter (300m) in the
lateral melting scheme (Steele, 1992), Exp-PFSD uses the equations described in Section 2.2
to determine the prognostic FSD and related physical processes (see Table 2). With the
prognostic FSD, the evolution of SIA in Exp-PFSD (Fig. 2a, red line) shows smaller SIA in the
melting months (June to September) and similar magnitude of SIA in other months compared
to that of Exp-CFSD (Fig. 2a, blue line) during 2016-2018. After that, Exp-PFSD simulates
smaller SIA than that of Exp-CFSD for most months during 2019-2020, especially for the
seasonal maximum of 2019 and SIA after May, 2020.
Figure 3 shows the evolution of sea ice mass budget terms with cell-area weighted
averaging over the entire model domain with 15-day running-average for smoothing out high-





frequency fluctuations for all experiments. The most notable difference between Exp-CFSD
and Exp-PFSD is the magnitude of basal melt (red lines) and lateral melt (grey lines). In Exp-
CFSD, basal melt plays the dominant role in reducing sea ice mass compared to lateral melt
that has negligible contribution to the total mass change. As discussed in Maykut and Perovich
(1987), the inclusion of friction velocity in calculating the lateral melting rate results in $w_{lat} \rightarrow$
0 as $u_* \rightarrow 0$, which contributes to negligible lateral melt in Exp-CFSD. By contrast, Exp-PFSD
with prognostic floe size shows that lateral melt has the major contribution in reducing ice mass
(Fig. 3b), a result of smaller floe size near the ice edge simulated by Exp-PFSD (Fig. 10a). It
is also notable that the increased lateral melt in Exp-PFSD tends to be compensated by the
decreased basal melt (Fig. 3b). The overall ice melt due to oceanic processes in Exp-PFSD (i.e.,
the sum of lateral melt and basal melt) does not change significantly compared to that of Exp-
CFSD (Fig. S2e). The melting potential in the CICE model of CAPS, the available energy from
the ocean to melt sea ice, is defined as the vertical integral of the difference between ocean
temperature and freezing point. When the available oceanic energy is less than the sum of heat
fluxes used for lateral and basal melt, the CICE model performs a linear scaling to maintain
the relative magnitude of heat fluxes for lateral and basal melt. Thus, the increased energy
consumption by lateral melt due to smaller floe size reduces the available energy for basal melt.
Such change between lateral and basal melt has been shown in some studies (e.g., Bateson et
al., 2020, 2022; Roach et al., 2018a, 2019; Smith et al., 2022; Tsamados et al., 2015). Although
the rough compensation, Exp-PFSD simulates more ice melted by the oceanic energy compared
to Exp-CFSD from January to July (Fig. S2e).



Figure 4 shows the evolution of ice-ocean heat flux, the friction velocity at the ice-ocean
interface, and the temperature difference between SST and freezing point for Exp-CFSD and
Exp-PFSD. These variables are the average of ice-covered cells with at least 1% ice
concentration, and the ice-ocean heat flux is weighted by the ice concentration so that the
weighted heat flux represents the mean value of cell, rather than the mean value of ice-ocean
interface. It should be noted that cells with negative values of the temperature difference (i.e.,
supercooled water) are forced to be zero. This is consistent with the treatment in the CICE
model for the calculation of ice-ocean heat flux. As shown in Fig. 4a and Fig. S2e, the evolution
of ocean-induced ice melt is consistent with that of the ice-ocean heat flux for both Exp-CFSD
and Exp-PFSD. Both Exp-CFSD and Exp-PFSD show relatively similar evolution of the
friction velocity (Fig. 4b). The temperature difference of Exp-PFSD is much smaller than that
of Exp-CFSD (Fig. 4c). The ice-ocean heat flux is the total heat flux from ocean to ice through
ice bottom surface and lateral surface. Although Exp-PFSD has smaller temperature difference
as well as the melting potential under ice-covered cells, the larger total ice surface area due to
smaller floe size increases the efficiency of Exp-PFSD extracting energy from the ocean. The
smaller temperature difference of Exp-PFSD and the compensation between lateral and basal
melt suggest that the ocean surface layer of Exp-PFSD is more closed to the freezing point
compared to that of Exp-CFSD. Energy loss from the ocean through air-sea heat flux that
further cools the upper ocean, freshwater input (e.g., ice melting, precipitation) that raises the
freezing point, as well as non-physical numerical oscillation (Naughten et al., 2018; Yang et
al., 2022) can lead to increased frazil ice formation of Exp-PFSD as shown in Fig. 3a-b and



Fig. S2g.

Figure 5 shows the heat flux budget at the ice surface averaged for all ice-covered cells.

The positive ice-atmosphere heat fluxes of Exp-CFSD and Exp-PFSD in July (Fig. S3a)
correspond to top melt in Fig. 3a-b and Fig. S2b (as well as Table S2). The ice-atmosphere heat
flux not only determines the magnitude of ice surface melt in summer but also the energy loss
from the ice interior in winter, which is crucial for the ice growth. As shown in Fig. S3a, Exp-
PFSD loses more energy to the atmosphere than that of Exp-CFSD in most winters. The
conductive heat flux also shows similar evolution, suggesting that more energy is conducted to
the ice top from ice layers below in Exp-PFSD (Fig. S3b). The loss of ice energy then
contributes to increased ice growth at the ice bottom as shown in Fig. 3a-b and Fig. S2f (as
well as Table S2). Generally, the net shortwave flux of Exp-PFSD is larger (ice gains more
energy) than that of Exp-CFSD, especially during the melting season (Fig. S3c). In contrast to
the net shortwave flux, for most of the time, the net longwave flux of Exp-PFSD is smaller (i.e.,
ice loses more energy) than that of Exp-CFSD (Fig. S3d). Exp-PFSD loses more energy
through sensible heat flux compared to Exp-CFSD (Fig. S3e). For latent heat flux, there is no
common features between Exp-PFSD and Exp-CFSD, suggesting the difference in the
simulation of atmospheric transient systems (Fig. S3f).

The ice mass budget in Fig. 3 is not directly related to the evolution of sea ice area in Fig.

2 since each process acts differently in changing ice area. For vertical processes (i.e., top melt,
basal melt), ice must be vertically-melted completely to reduce ice area. Lateral melt, on the
contrary, can directly reduce ice area (Smith et al., 2022). Figure 6 shows the evolution of sea



ice area changes due to thermal processes (top melt, basal melt, lateral melt, frazil ice formation)
and dynamical processes (transport, ridging). For thermal area changes, Exp-PFSD (red line),
in general, shows comparable ice area changes to increased ice area compared to Exp-CFSD
(blue line) for most of the period (Fig. 6a). Compared with Fig. S2g, the timings that Exp-
PFSD shows more thermally-increased ice area correspond to increased frazil ice formation,
which primarily occurs in open water. In contrast to thermal area changes, dynamical area
changes of Exp-PFSD tends to reduce ice area relative to that of Exp-CFSD (Fig. 6e).
Dynamically-induced area changes are partly due to the ridging scheme (Lipscomb et al., 2007)
that favors the conversion of thin ice to thicker ice and reduces total ice area but preserves the
total volume. In general, Exp-PFSD has higher fraction of ice in the thinner ITD range than
Exp-CFSD.
Based on geographic features, we define the following subregions for further analysis: 1)
Barents and Greenland Seas (ATL, 45W-60E, 65N-85N), 2) Laptev and Kara Seas (LK, 60E-
150E, 65N-85N), and 3) Beaufort, Chukchi, and East Siberian Seas (BCE, 150E-120W, 65N-
85N, see black boxes in Fig. 1 for the geographic coverage of subregions). The fetch of ATL,
LK and BCE regions are limited by the surrounding continents and the seasonal evolution of
ice-covered area. The ATL region is only partially-limited by ice-covered area while the LK
and BCE regions can be fully-covered by sea ice in winters. Figure 7 shows the evolution of
sea ice extent, sea ice area, domain-averaged significant wave height, melting potential, and
heat flux at the ocean surface (FLUX$_{OCN}$, including ice-ocean and atmosphere-ocean interfaces)
of Exp-CFSD and Exp-PFSD. As shown in Fig. 7a-c, it is clear that the higher (lower)



significant wave height corresponds to less (more) regional ice coverage for all subregions. For
the melting potential (Fig. 7d), the difference between Exp-CFSD (blue line) and Exp-PFSD
(red line) in August, in general, is correlated with $FLUX_{OCN}$ in July (Fig. 7e). The more (less)
incoming heat flux to the ocean due to less (more) ice-covered area increases (decreases)
energy stored in the ocean surface layer. However, $FLUX_{OCN}$ alone cannot explain the
difference of the melting potential for the entire period. For example, Exp-PFSD shows more
melting potential after December, 2019 in ATL region (Fig. $d_1$), and more melting potential in
December, 2017 in LK region (Fig. $d_2$) compared to Exp-CFSD. These timings do not show
corresponded $FLUX_{OCN}$ at the preceding month, suggesting the contribution of different
processes. Figure 8 shows the evolution of wave energy dissipation due to whitecapping and
the difference of temperature profile in the upper 150m for Exp-CFSD and Exp-PFSD. As
described in section 3, wave energy dissipation increases the turbulent kinetic energy in the
surface layer and thus vertical mixing. Dissipation due to surface wave breaking is zero for
most of the period. Occasionally, there are non-zero dissipations due to surface wave breaking
for Exp-CFSD and Exp-PFSD. The evolution of wave dissipation due to whitecapping (Fig.
8a-c) is in good agreement with that of significant wave height in Fig. 7c. This suggests that
stronger wave conditions associated with less ice-covered area increase the effect of vertical
mixing. Combined with the warmer upper ocean in Exp-PFSD after January, 2020 in ATL
region and in December, 2017 in LK region in Fig. 8d-e, the strengthened vertical mixing
brings warmer water of the subsurface upward and maintains/increases the melting potential in
the subregions.



Additionally, atmospheric circulation responds to the changes in spatial distribution of sea
ice (Fig. S1). As shown in Figure S4, Exp-PFSD tends to have anomalous anti-cyclonic
circulations in September compared to Exp-CFSD, but there is no consistent center of actions
during the entire period. In March, Exp-PFSD tends to simulate anomalous cyclonic
circulations in the Barents-Kara Sea for most of the years compared to Exp-CFSD, except 2019.
The response in the atmospheric state in both experiments also influence sea ice movement,
which further contributes to the regional ice differences in Fig. 7a-b, as well as the heat flux
budgets in Fig. S3.
**4.2  Sensitivity to lateral melting rate parameterization**
In addition to the floe size as discussed in the above section, the lateral melting rate ($w_{lat}$)
is an important factor contributing the relative strength of lateral and basal melt. As described
in section 3, we conduct the experiments with the lateral melting rate suggested by Perovich
(1983, P83), and Maykut and Perovich (1987, MP87) (see Table 2), to examine the sensitivity
of Arctic sea ice simulation to different lateral melting rate parameterizations. As shown in Fig.
2b, the simulated summer sea ice area of Exp-LatMelt-C (green line) and Exp-LatMelt-P (grey
line), in general, is larger than those of Exp-CFSD (blue line) and Exp-PFSD (red line).
As shown in sea ice mass budget (Fig. 3a, 3c), Exp-LatMelt-C, which does not include
the friction velocity in the formulation (Equ. 23), but keeps other model configurations same
as Exp-CFSD only show slightly larger contribution to lateral melt during summer months (Fig.
S5d). Also, the contribution to basal melt by Exp-LatMelt-C is generally smaller than that in
Exp-CFSD (Fig. S5c). Similar to the experiments with MP87 scheme, Exp-LatMelt-P with the





prognostic FSD also shows the compensation between lateral melt and basal melt compared to
Exp-LatMelt-C (Fig. 3c, 3d). Exp-LatMelt-P show stronger lateral melt compared to Exp-
PFSD, which is contributed by the P83 formulation (Fig. S5d). Despite the stronger lateral melt
in Exp-LatMelt-P, its basal melt is smaller compared to Exp-PFSD (Fig. S5c). Thus, the ocean-
induced melt of Exp-LatMelt-P is broadly similar to that of Exp-PFSD. The result of Exp-
LatMelt-P and Exp-PFSD suggests that the changes of lateral and basal melt due to different
lateral melting rate parameterizations are mostly controlled by the available energy from the
ocean (i.e., melting potential).

Exp-LatMelt-P simulates more basal growth in winter (Fig. S5f), which is contributed by

more energy loss to the atmosphere (Fig. 5a), in comparison to Exp-PFSD. Also, more frazil
ice formation is simulated in Exp-LatMelt-P than Exp-PFSD during most of simulation period
(Fig. S5g). The combined effects of above processes lead to that Exp-LatMelt-P shows less
total ice melt in summer and similar ice growth in winter compared to Exp-PFSD (Fig. S5a).
Due to more frazil ice formation, Exp-LatMelt-P shows more thermally-increased ice area
compared to Exp-PFSD (Fig. 6, Fig. S5g). Frazil ice formation reduces open-water areas and
blocks the energy exchange between the atmosphere and the ocean. That is, the upper ocean
under sea ice in Exp-LatMelt-P receives less incoming flux from the atmosphere (i.e., solar
radiation) during April to September (not shown) to balance the energy consumption by ice
melt, which leads to smaller ocean temperature difference compared to Exp-PFSD (Fig. 4c,
green and red lines).

Figure 9 shows the spatial distribution of sea ice concentration, sea surface temperature,

and friction velocity in September, 2020 for the experiments using MP87 and P83 schemes.
Exp-CFSD, Exp-PFSD, and Exp-LatMelt-C simulate large areas with ice concentration less
than 5% (they are mostly much less than 1%, Fig. 9a$_{1-3}$). In opposite to these three experiments,
Exp-LatMelt-P does not show wide areas with non-zero and infinitesimal ice concentration
(Fig. 9a$_4$). Although these areas only account for a tiny fraction of total sea ice, they may still
be a source of uncertainty for sea ice simulations. Ice-existed cells can be influenced by all
processes involved in sea ice mass budget (Fig. 3) while ice-free cells can only be affected by
frazil ice formation and dynamical advection. Under these small-ice areas, SST is well above
the freezing point (Fig. 9b) and the friction velocity is mostly less than $5 \times 10^{-4}$ $m/s$ (Fig.
9c). In our configuration of CICE model, the minimum value of friction velocity is set to
$5 \times 10^{-4}$ $m/s$. This suggests that the friction velocity is the limit factor for heat flux
transferred into sea ice in the small-ice areas. For basal heat flux, the formulation in the CICE
model is based on Maykut and McPhee (1995), which is controlled by the friction velocity and
the temperature difference. Thus, basal heat fluxes with small friction velocities may not be
large enough to satisfy the energy convergence (in conjunction with conductive heat flux at the
ice bottom) at the ice-ocean interface to melt ice if the temperature difference does not show
larger magnitude. Since MP87 lateral melting scheme includes the friction velocity, lateral heat
flux is also limited in small-ice areas. Exp-PFSD has much smaller floe size (compared to
300m) in these small-ice areas, but the increased strength of lateral melt does not overcome the
limitation of friction velocity to melt ice completely (Fig. 9a$_2$). The P83 lateral melting scheme
that does not include the friction velocity is controlled by the temperature difference, but the



effect of lateral melting in Exp-LatMelt-C is largely constrained by constant 300m floe
diameter. Liang et al. (2019) suggested theses small-ice areas can be eliminated by assimilating
SST observations. The results of Exp-LatMelt-P suggest a model physic approach that
considers the prognostic FSD and the lateral melting rate to reduce the coverage of small-ice
near the ice-edge.
**4.3 Sensitivity to floe-fracturing parameterization**
The equally-redistributed formulation (hereafter PF1) for floe-fracturing described in
section 2.3.a does not have preferential floe size after fracturing (i.e., a stochastic process).
However, the size of fractured floes can be predicted based on the properties of surface ocean
waves, particularly wavelength (Dumont et al. 2011; Horvat and Tziperman, 2015). In this
section, we conduct an experiment (Exp-WaveFrac-P, see Table 2), which utilizes a semi-
empirical wave spectrum to redistribute fractured floes (see section 2.3.b for details and
hereafter PF2) to explore the effects of different wave-fracturing formulations on Arctic sea ice
simulation. As shown in Fig. 2c, Exp-WaveFrac-P (orange line) simulates larger ice area in
summer and comparable ice area in winter with respect to Exp-PFSD (red line).
By applying different formulations for floe-fracturing (as well as different lateral melting
rate formulations), the FSD responds accordingly. To quantify the responses of FSD associated
with different physical configurations (Table 2), the representative floe radius $r_a$, as well as its
tendency due to different processes in the equation (9) are utilized and calculated as (Roach et
al., 2018a),



$$r_a = \frac{\int_{\mathcal{R}} \int_{\mathcal{H}} r f(r,h) dr dh}{\int_{\mathcal{R}} \int_{\mathcal{H}} f(r,h) dr dh} \quad (25)$$

$$\frac{dr_a}{dt} = \frac{\int_{\mathcal{R}} \int_{\mathcal{H}} r \frac{df(r,h)}{dt} dr dh}{\int_{\mathcal{R}} \int_{\mathcal{H}} f(r,h) dr dh} \quad (26)$$

Figure 10 shows the spatial distribution of the representative floe radius in winter and
summer for all experiments with the prognostic FSD. As described in section 3, $L(r,h)$ is
initialized by the power law distribution with the exponent as 2.1 for all experiments. Exp-
WaveFrac-P shows smaller floe radius in the Chukchi and East Siberian Seas and north of
Greenland at the early stage of simulation compared to experiments using PF1 formulation (Fig.
$10a_1$-$c_1$, upper panel). Small-floe areas in Exp-WaveFrac-P are mostly contributed by the effect
of wave-fracturing where decreasing tendency of floe radius can extend further into the central
Arctic from the Atlantic and the Bering Strait compared to PF1 experiments (Fig. S6). After
September, 2016, the representative floe radii of PF experiments emerge, that is, Exp-
WaveFrac-P has smaller floe size compared to PF1 experiments for both winter and summer
(Fig. 10a-c). In summer, Exp-WaveFrac-P shows mostly fully-fractured floe (<10m, Fig. 10c,
bottom panel). The stronger wave-fracturing shown in Exp-WaveFrac-P is partly contributed
by the semi-empirical wave spectrum used in PF2. The simulated wave parameters under ice-
covered area are mostly with $H_s < 0.01\ m/s$ and $T_p > 15\ s$. The constructed wave spectrum
and amplitude based on simulated wave parameters under sea ice and equations (17) and (18)
still include the contribution from high-frequency waves ($T = 2s\ bin$), especially in the ice
pack far from the ice edge. The high-frequency waves only account for small fraction in the
wave period distribution $\overline{P}(T)$, and have small wave amplitude $A(T)$ ($\sim 7 \times 10^{-4} m$). The





strain of high-frequency bin based on equation (20) still exceeds the yielding strain and then
fractures floe into the smallest floe size category. Observational and numerical studies showed
that high-frequency waves rapidly decay and reach "zero" transmission state for high-
frequency waves when traveling under sea ice (e.g., Collins et al., 2015; Liu et al., 2020).
Despite the over-fracturing behavior shown in Exp-WaveFrac-P, the prevalence of small-floe
does not translate into the stronger ocean-induced ice melt but weaker melt in summer
compared to Exp-PFSD (Fig. 3d-e, Fig. S7e), indicating the limiting role of melting potential.
The weaker ocean-induced ice melt in summer of Exp-WaveFrac-P is corresponded to smaller
ice-ocean heat fluxes (Fig. S8a), which is contributed by both smaller friction velocity and
temperature difference (Fig. S8b-c).
**4.4  Sensitivity to wave-attenuation parameterization**
We have shown that ocean waves can alter the upper ocean through wave-enhanced
mixing, which may affect sea ice locally (Fig. 8, see section 4.1). The results from PF1 and
PF2 experiments imply that the simulated wave parameters can determine how ice floes are
fractured. As described in section 2.1, we can choose different coefficients in equation (2) to
control the wave attenuation rate of each frequency. In this section, we conduct experiments
using R18 coefficients (see section 3 and Table 2) to study the impacts of wave-attenuation rate
on Arctic sea ice simulation. The simulated sea ice area in Exp-WaveAtt-C (Fig. 2d, light-blue
line) resembles that in Exp-CFSD (Fig. 2d, blue line) before 2019. After 2019, Exp-WaveAtt-
C simulates smaller ice area compared to Exp-CFSD. Since both Exp-CFSD and Exp-WaveAtt-
C use constant floe size, which allows us to neglect the effect of spatial distribution of floe size





and MP87 lateral melting rate, which make lateral melt have negligible contribution (Fig. S9d),
basal melt is the primary factor for the ocean-induced ice melt during the entire period (Fig. 3a,
3f, and Fig. S9e). The strength of basal melt in Exp-WaveAtt-C is weaker than that in Exp-
CFSD from April, 2018 to January, 2020 (Fig. S9c). Basal growth of Exp-WaveAtt-C is also
smaller than that of Exp-CFSD in the winter of 2018 and 2019 (Fig. S9f). Compared to Exp-
CFSD, Exp-WaveAtt-C shows stronger top melt in summer of 2018 (Fig. S9b). The combined
effects of above processes lead to thinner ice state in Exp-WaveAtt-C before 2019 (Fig. S9a).
The thinner state of Exp-WaveAtt-C in the winter of 2019 make more open-water be created
by basal melt (regardless of its smaller magnitude) and thus smaller SIA (Fig. 2d), which is
also shown in the thermally-induced ice area changes that Exp-WaveAtt-C has smaller
magnitude in the corresponded period (Fig. 6d). As discussed in section 4.1, top melt and basal
growth is in good agreement with the ice-atmosphere heat flux (Fig. S9, S10a). That is, ice
mass and area changes described above are mainly driven by the ice-atmosphere heat flux
associated with the atmospheric responses to the changes in ocean wave conditions.

Different from the M14 experiments, the simulated sea ice area of Exp-WaveAtt-C (light-

blue line) and Exp-WaveAtt-P (yellow line) show relatively similar evolution during 2016-
2020 (Fig. 2d). The R18 coefficients represent weaker wave attenuation relative to the M14
coefficients. Thus, ocean waves in the R18 experiments are expected to transmit further into
the ice pack while maintaining relatively higher wave energy. To quantify to what extent the
ice can be affected by wave, we calculate the wave-affected extent (WAE), which is defined as
the sum of the area of cells with ice concentration greater than 15% and significant wave height





greater than 30cm (Cooper et al., 2022). Figure 11 shows the evolution of WAE for the M14
and R18 experiments with 15-day running average to smooth the high-frequency changes of
wave conditions. The weaker attenuation in Exp-WaveAtt-C and Exp-WaveAtt-P results in
generally larger WAE compared to Exp-CFSD and Exp-PFSD (as well as all previous
experiments with M14 coefficients, not shown). The direct impact of larger WAE in Exp-
WaveAtt-P is that the representative floe radius is mostly smaller than 10m (fully-fractured by
waves) (Fig. 10d). The decreasing tendency of floe radius due to wave-fracturing is the
dominant factor contributed to the fully-fractured condition (Fig. S6). Similar to Exp-
WaveFrac-P, the fully-fractured condition does not lead to stronger ocean-induced melt due to
limited oceanic energy (Fig. 3b, 3e, 3g, S9e).
**5.    Conclusions and Discussions**

This study investigates the impacts of ocean waves on Arctic sea ice simulation based on

a newly-developed atmosphere-ocean-wave-sea ice coupled model, which is built on the
Coupled Arctic Prediction System (CAPS) by coupling the Simulating Waves Nearshore
(SWAN) and the implementation of the modified joint floe size and thickness distribution
(FSTD). A set of pan-Arctic experiments with different configurations of FSD-related
processes are performed for the period 2016-2020. Specifically, we examine the contrasting
behaviors of sea ice between constant and prognostic floe size, the responses of sea ice to
different lateral melting rate formulations, and the sensitivity of sea ice to the simulated wave
parameters under the atmosphere-ocean-wave-sea ice coupled framework.

The results of FSD-fixed and FSD-varied experiments show that the simulated sea ice





area is generally lower with smaller floe size associated with physical processes that change
FSD. According to sea ice mass budget analysis, smaller floe size contributes to increased
lateral melt, but its effect is reduced by decreased basal melt. The combined effects of lateral
and basal melt associated with smaller floe size result in relatively more ice melt by the ocean
energy, which is similar to previous studies (e.g., Bateson et al., 2022; Roach et al., 2019; Smith
et al., 2022). The simulations in Smith et al. (2022) with varying lateral melting strength based
on the Community Earth System Model version 2 (CESM2) with a slab-ocean model showed
minimal change in frazil ice formation. In our simulation with a full ocean model, the enhanced
ice melt by the ocean, though it is partially balanced by increased frazil ice formation due to
the depletion of melting potential in the surface layer. This suggests negative feedback from
the full ocean physics. Our fully-coupled simulations also show that atmospheric states respond
to changing ice distributions and then modify the energy budget at the ice surface that
determines top melt in summer and basal growth in winter. The FSD-varied experiments, in
general, show more energy loss from ice to atmosphere in winter, and all experiments show
year-to-year variations of energy gain for sea ice in summer.
The depletion of ocean energy in the surface layer as well as enhanced frazil ice formation
are the direct responses to the changes of ice-ocean coupling with the prognostic FSD. The
fractured sea ice enlarges the ice-ocean heat flux while the freezing temperature is still
determined by the sea surface salinity in the ocean model. However, the local salinity at the
ice-ocean interface can be significantly lower than sea surface salinity, and thus higher freezing
temperature locally due to the meltwater from sea ice (e.g., the false-bottom, Notz et al., 2003).





Schmidt et al. (2004) proposed the ice-ocean heat flux formulation that considers the local
salinity equilibrium but its formulation is only for the ice bottom interface. The generalization
of ice-ocean heat flux with the consideration of local salinity equilibrium for both bottom and
lateral interface might yield a more realistic ice-ocean coupled simulation. Although the lateral
melting rate formulation does not have the major effect on the simulated floe size distribution,
the simulated sea ice area and ice mass budget are sensitive to the choice of the formulation.
The lateral melting rate formulations applied in this study as well as previous laboratory results
are not related the ice properties (i.e., ice thickness and floe size, Josberger and Martin, 1981;
Maykut and Perovich, 1987; Perovich, 1983). A recent laboratory study suggested that the
lateral melting rate is a function of temperature difference and the ratio of floe size to ice
thickness (Li et al., 2021). Smith et al. (2022) also suggested that Arctic sea ice simulation can
be sensitive to the lateral melting rate of Perovich (1983) with different weights on each ice
thickness category. Further studies are required to investigate improved lateral melting rate
parameterization with observational constraints (e.g., data from the MOSAiC campaign in
2020, Nicolaus et al., 2021) within the prognostic FSD framework.
As discussed in Horvat and Tziperman (2015), the FTSD is sensitive to the wave
attenuation coefficients. Our simulations also show substantially contrasting behaviors in the
simulated floe size distribution associated with simulated wave parameters, suggesting that
several aspects need further investigation. First, the empirical wave attenuation (i.e., IC4M2)
may have reasonable performance in simulating the changes of wave energy spectrum locally
with specific ice conditions (e.g., Liu et al., 2020). However, the dissipation of wave energy



varies spatially for the pan-Arctic (as well as pan-Antarctic) scale simulation with the different
sea ice properties (i.e., ice concentration, ice thickness, floe size). Thus, a viscous boundary
layer model (Liu et al., 1991) or a viscoelastic model (Wang and Shen, 2010) for wave
attenuation, which provides spatially-varied wave attenuation with respect to sea ice properties,
might be able to give more realistic simulations in the wave-fracturing process and thus the
floe size distribution. Also, the current implementation of sea ice effects in the SWAN model
does not include the reflection and scattering due to sea ice, which redistributes the wave energy
spatially and potentially changes the wave-fracturing behavior. Second, the probability of floe-
fracturing $Q(r)$ in both formulations applied in this study are uncertain. Both formulations
result in floe-fracturing into smaller floe size categories within a short time-interval as long as
the simulated wave parameters satisfying the yielding strain. This strong contribution in the
wave-fracturing term is not easily balanced by the floe-welding term. The floe-welding term
(Roach et al., 2018a, b) acts to reduce the floe number density so that it is less effective in
increasing the representative floe radius if the floe is mostly fractured with the smallest floe
size. Third, the attenuated wave energy by sea ice does not influence sea ice conditions in this
study. As suggested by Longuet-Higgins and Steward (1962), the attenuated wave energy is
transferred into the ocean (as we described in section 3 for wave-enhanced mixing) or sea ice.
For sea ice, the transferred energy acts as a stress, called wave radiation stress (WRS), pushing
sea ice to the direction of wave propagation. By including the WRS in the momentum equation
of ice, the WRS then can affect sea ice drift (e.g., Boutin et al., 2020).

For quantitative applications (e.g., forecasting sea ice), more observations (especially



ocean waves under sea ice and FSD) are needed to reduce uncertainties in the atmosphere-
ocean-wave-sea ice coupled model, particular wave-related processes in ice-covered regions.
Horvat et al. (2019) developed a new technique to retrieve pan-Arctic scale FSD climatology
and seasonal cycle from CryoSat-2 radar altimeter and this method can resolve floe size from
300 m to 100 km and potentially up to 20 m scale if applying to ICESat-2 data. ICESat-2
altimetry also provides a new opportunity to observe ocean waves in sea ice at hemispheric-
scale coverage by directly observing the vertical displacements of the ice surface (e.g., Horvat
et al., 2020). In situ observations, despite their limited spatial coverage, are valuable wave
spectra measurements for wave-physics validation and improvement (e.g., Cooper et al., 2022;
Liu et al., 2020).



Code and data availability: The outputs of pan-Arctic simulation analyzed in this study
are archived in https://doi.org/10.5281/zenodo.7922725.

Author contributions: CYY and JL designed the model experiments, developed the
updated CAPS model, and wrote the manuscript, CYY conducted the experiments and analyzed
the results. DC provided constructive feedback on the manuscript.

Competing interests: The authors declare that they have no conflict of interest.

Acknowledgements: This research is supported by the National Natural Science
Foundation of China (42006188), the National Key R&D Program of China
(2018YFA0605901), and the Innovation Group Project of Southern Marine Science and
Engineering Guangdong Laboratory (Zhuhai) (311021008).



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

Table 1 The summary of physic parameterizations used in all pan-Arctic simulations.

| WRF physics | |
|---|---|
| Cumulus | Grell-Freitas (Freitas et al. 2018) |
| Microphysics | Morrison 2-moment (Morrison et al. 2009) |
| Longwave radiation | CAM spectral band scheme (Collins et al. 2004) |
| Shortwave radiation | CAM spectral band scheme (Collins et al. 2004) |
| Boundary layer | MYNN (Nakanishi and Niino, 2009) |
| Land surface | Unified Noah LSM (Chen and Dudhia, 2001) |
| | |
| ROMS physics | |
| Tracer advection | Upwind third-order horizontal advection (U3H; Shchepetkin, and McWilliams, 2005) Centered fourth-order vertical advection (C4V; Shchepetkin, and McWilliams, 2005) |
| Tracer vertical mixing | Generic Length-Scale scheme (Umlauf and Burchard, 2003) |
| | |
| CICE physics | |
| Ice dynamics | EVP (Hunke and Dukowicz, 1997) |
| Ice thermodynamics | Bitz and Lipscomb (1999) |
| Shortwave albedo | Delta-Eddington (Briegleb and Light, 2007) |
| | |
| SWAN physics | |
| Exponential wind growth | Komen et al. (1984) |
| Whitecapping | Komen et al. (1984) |
| Quadruplets | Hasselmann et al. (1985) |
| Depth-induced breaking | Battjes and Janssen (1978) |
| Bottom friction | Madsen et al. (1988) |
| Sea ice dissipation | Collins and Rogers (2017); Rogers (2019) |




Table 2 The summary of the experiments conducted in this study and their main changes in the experiment design. MP87: Maykut and Perovich (1987). P83: Perovich (1983). M14: Meylan et al. (2014). R18: Rogers et al. (2018).

| Experiment | Floe size | Lateral melting rate | Wave fracturing formulation | Wave attenuation coefficients |
|---|---|---|---|---|
| Exp-CFSD | Const. 300m | MP87 | None | M14 |
| Exp-PFSD | FSTD | MP87 | Equally (PF1) | M14 |
| Exp-LatMelt-C | Const. 300m | P83 | None | M14 |
| Exp-LatMelt-P | FSTD | P83 | Equally (PF1) | M14 |
| Exp-WaveFrac-P | FSTD | MP87 | Bretschneider (PF2) | M14 |
| Exp-WaveAtt-C | Const. 300m | MP87 | None | R18 |
| Exp-WaveAtt-P | FSTD | MP87 | Equally (PF1) | R18 |


**8.    Figures**

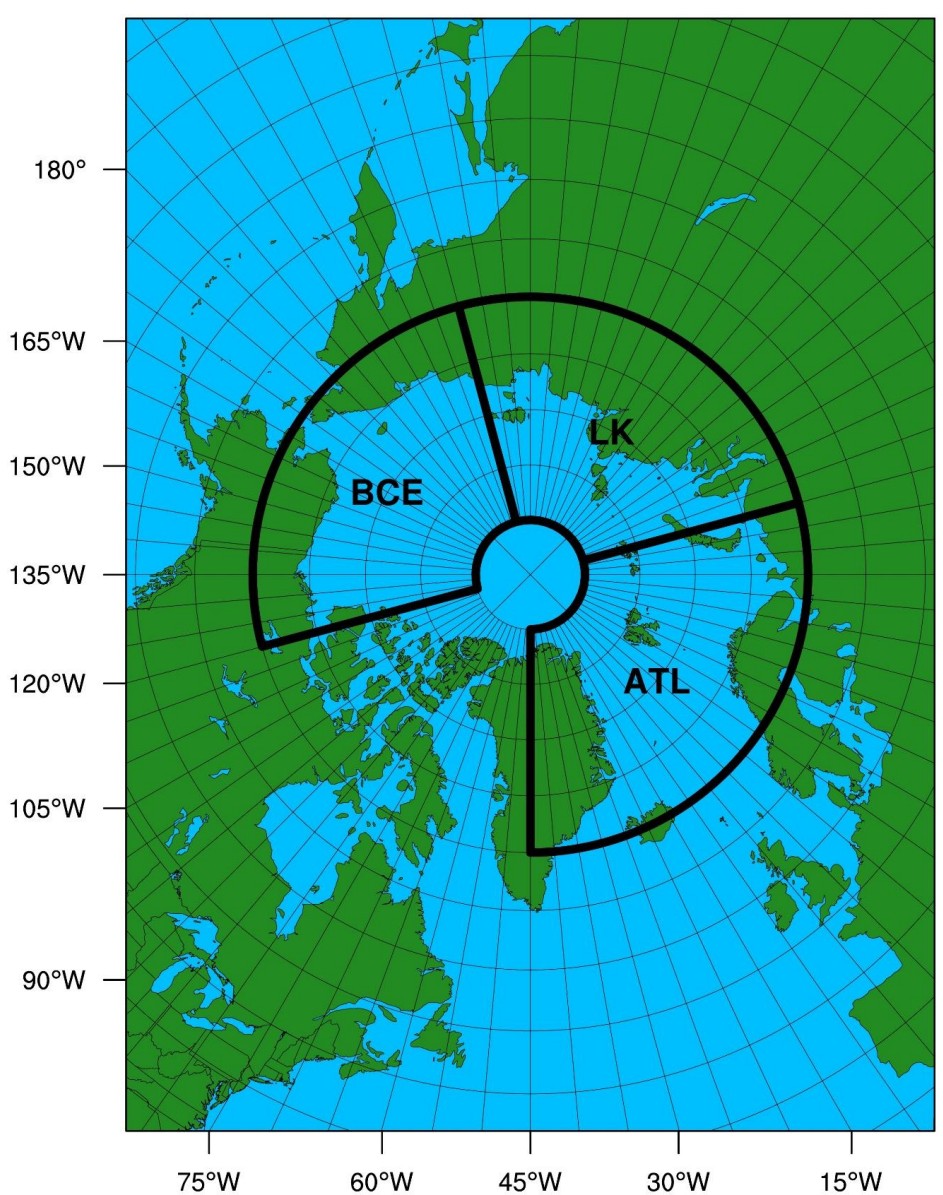


Figure 1 The model domain used in CAPS for pan-Arctic sea ice simulations. Black boxes
indicate the subregions for analysis performed in this study.


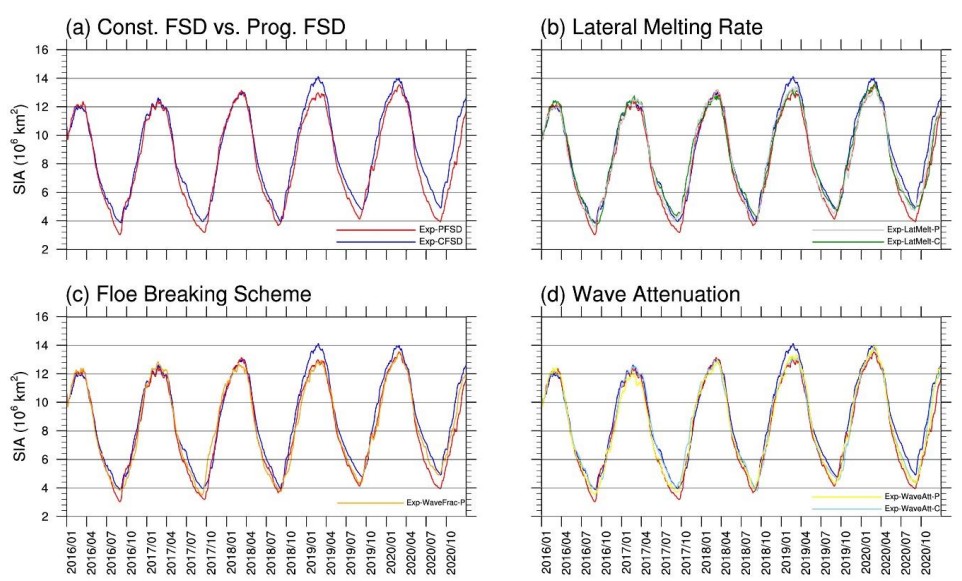


Figure 2 Time-series of Arctic sea ice area for Exp-CFSD (blue line), Exp-PFSD (red line),
Exp-LatMelt-C (green line), Exp-LatMelt-P (grey line), Exp-WaveFrac-P (orange line), Exp-
WaveAtt-C (light-blue line) and Exp-WaveAtt-P (yellow line).




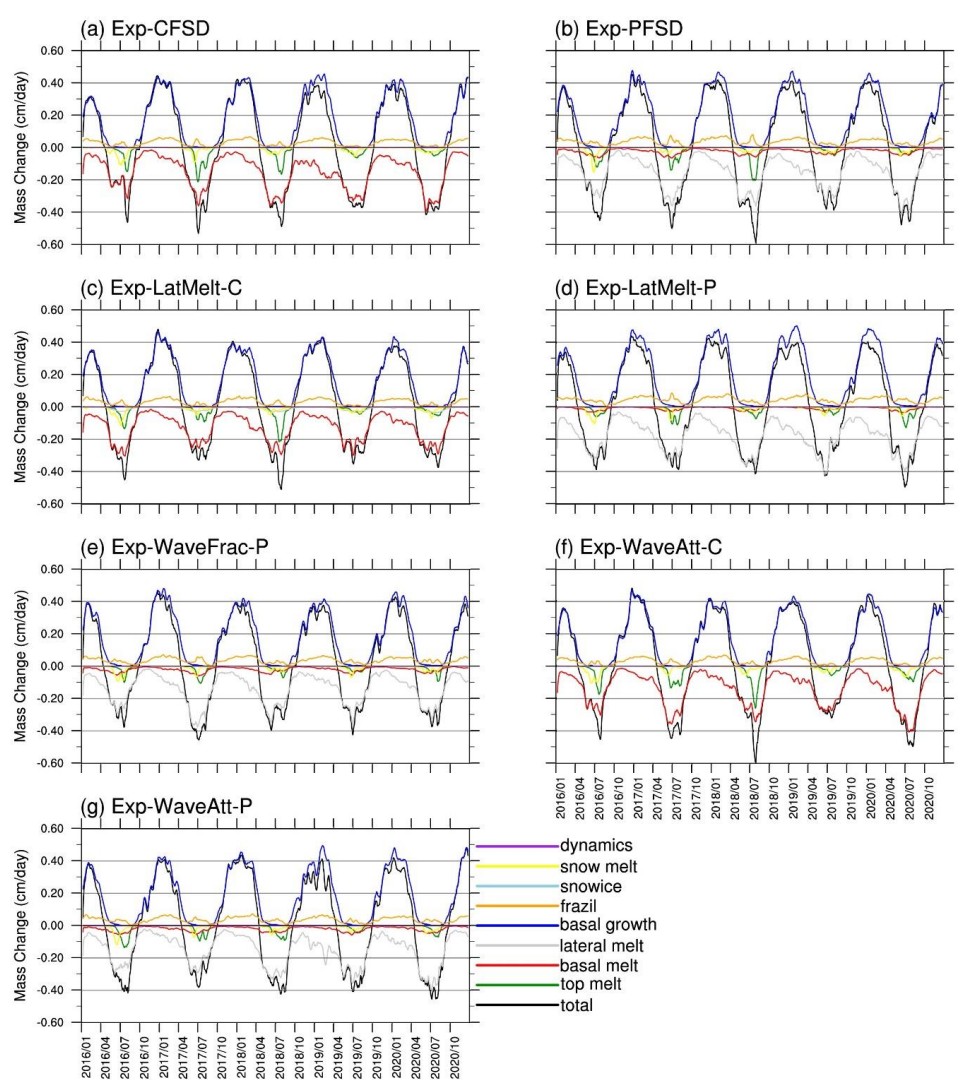

Figure 3 Time-series (15-day running-averaged) of sea ice mass budget terms for (a) Exp-CFSD, (b) Exp-PFSD, (c) Exp-LatMelt-C, (d) Exp-LatMelt-P, (e) Exp-WaveFrac-P, (f) Exp-WaveAtt-C, and (g) Exp-WaveAtt-P. Ice mass budget terms include: total mass change (black line), sea ice melt at the air-ice interface (top melt, green line), sea ice melt at the bottom of the ice (basal melt, red line), sea ice melt at the sides of the ice (lateral melt, grey line), sea ice growth at the bottom of the ice (basal growth, blue line), sea ice growth by supercooled open water (frazil, orange line), sea ice growth due to transformation of snow to sea ice (snowice, light-blue line), and sea ice mass change due to dynamics-related processes (dynamics, purple line) (Notz et al., 2016; Yang et al., 2022). For reference, snow melt term (yellow line) is included.





1080

Figure 4 Time-series (15-day running-averaged) of (a) ice-ocean heat flux, (b) friction velocity
at ice-ocean interface, and (c) the temperature difference between SST and freezing point for
Exp-CFSD (blue line), Exp-PFSD (red line), Exp-LatMelt-C (green line), and Exp-LatMelt-P
(grey line). Note: (a) is positive downward and weighted by ice concentration.

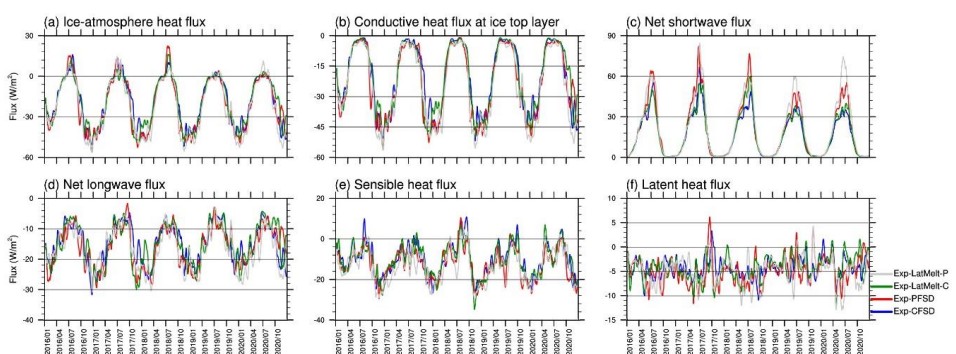

Figure 5 Time-series (15-day running-averaged) of (a) ice-atmosphere heat flux, (b) conductive heat flux at the ice top layer, (c) net shortwave flux, (d) net longwave flux, (e) sensible heat flux, and (f) latent heat flux for Exp-CFSD (blue line), Exp-PFSD (red line), Exp-LatMelt-C (green line), and Exp-LatMelt-P (grey line). Note: (a)-(e) are positive downwards and weighted by ice concentration.




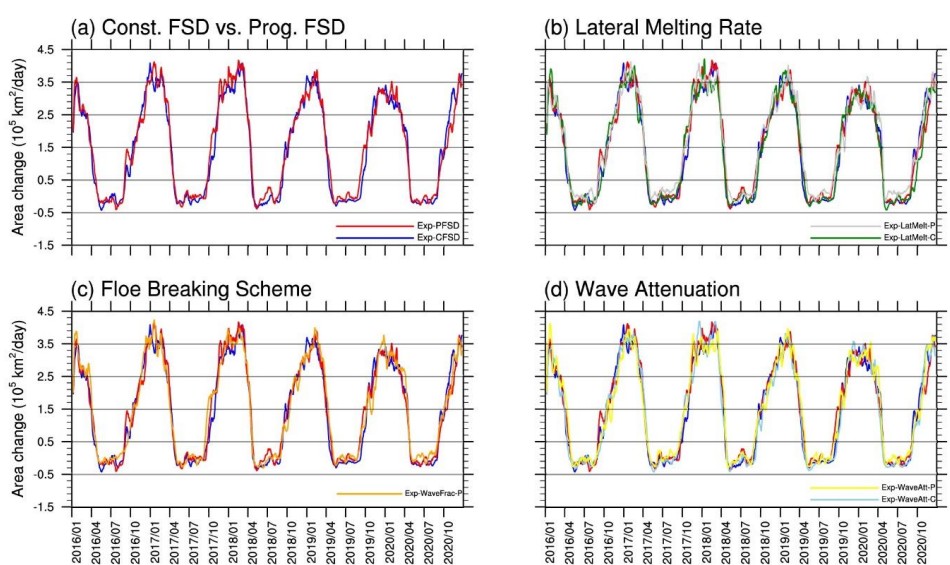

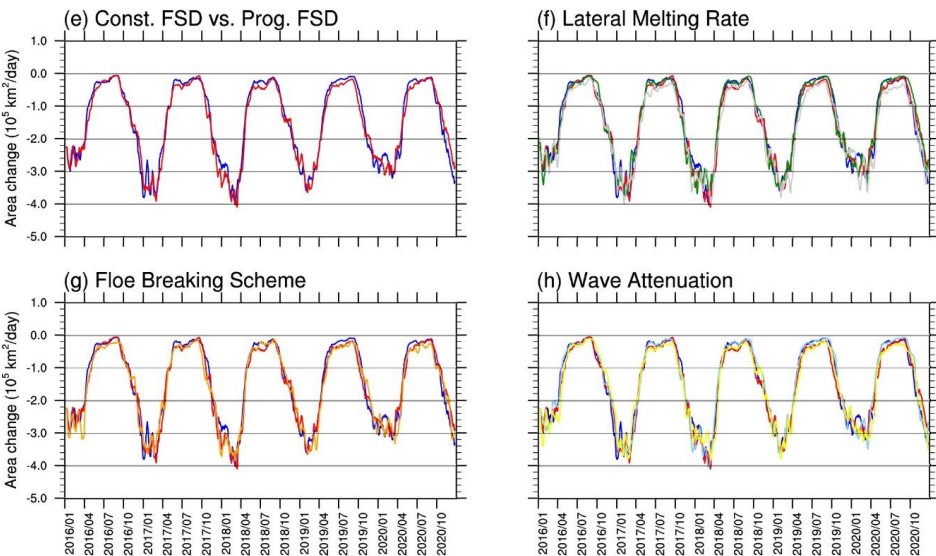

1092

Figure 6 Time-series (15-day running-averaged) of sea ice area changes due to thermal
processes (a-d, upper panel) and dynamical processes (e-h, bottom panel) for Exp-CFSD (blue
line), Exp-PFSD (red line), Exp-LatMelt-C (green line), Exp-LatMelt-P (grey line), Exp-
WaveFrac-P (orange line), Exp-WaveAtt-C (light-blue line) and Exp-WaveAtt-P (yellow line).


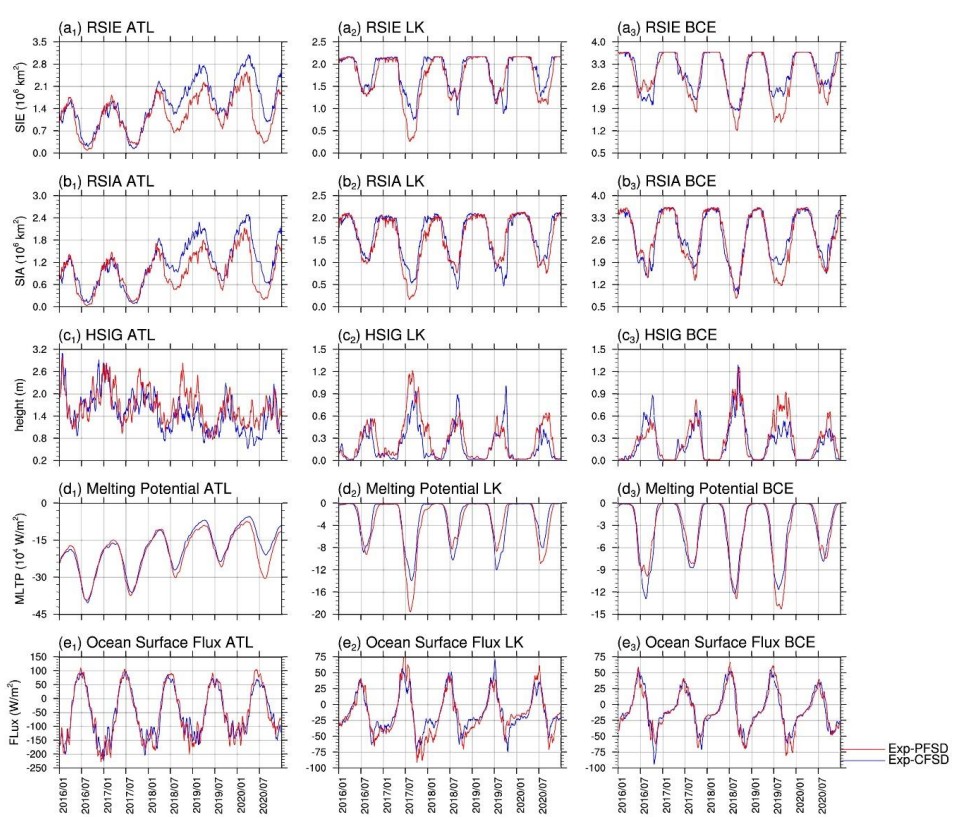

Figure 7 Time-series of (a) ice extent, (b) ice area, (c) significant wave height, (d) melting
potential, and (e) heat flux at the ocean surface in (1) ATL, (2) LK, and (3) BCE regions for
Exp-CFSD (blue line) and Exp-PFSD (red line). Note: (c)-(e) are region-averaged and 15-day
running-averaged values.




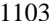

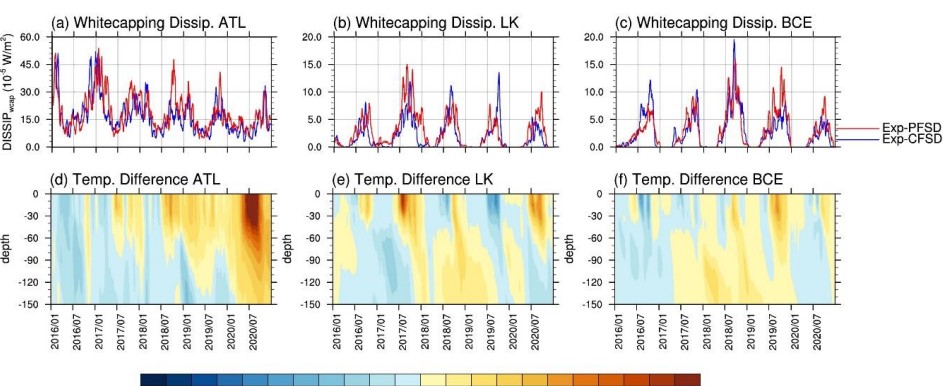


Figure 8 Time-series (15-day running-averaged) of white capping dissipation averaged over (a)
ATL, (b) LK, and (c) BCE regions for Exp-CFSD (blue line) and Exp-PFSD (red line), and the
temperature profile difference between Exp-CFSD and Exp-PFSD in the upper 150 m averaged
over (d) ATL, (e) LK, and (f) BCE regions.






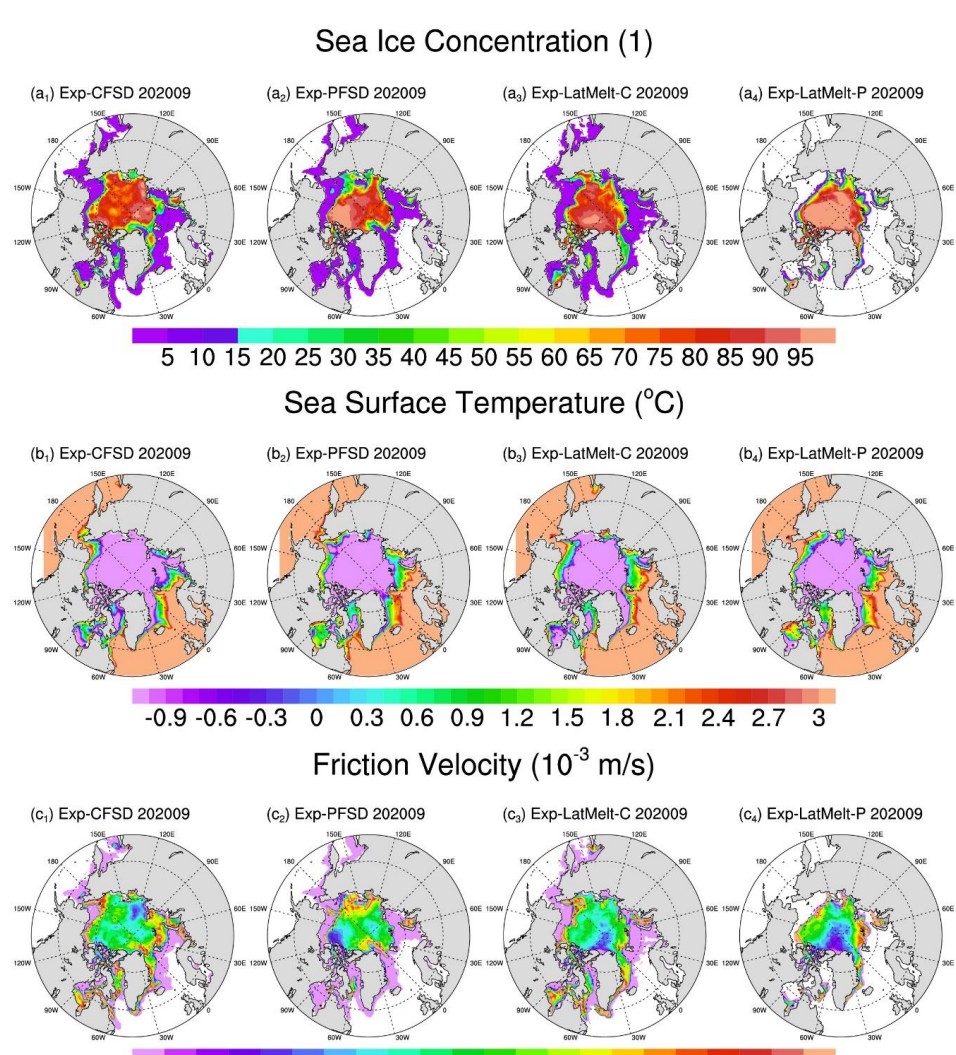


Figure 9 The monthly-mean of (a) sea ice concentration, (b) sea surface temperature, and (c)
friction velocity in September, 2020 for (1) Exp-CFSD, (2) Exp-PFSD, (3) Exp-LatMelt-C,
and (4) Exp-LatMelt-P.









Figure 10 The spatial distribution of the representative floe radius in March (upper panel) and September (bottom panel) of (a) Exp-PFSD, (b) Exp-LatMelt-P, (c) Exp-WaveFrac-P, and (d) Exp-WaveAtt-P for 2016-2020. Note: cells with less than 15% ice concentration are treated as missing values.

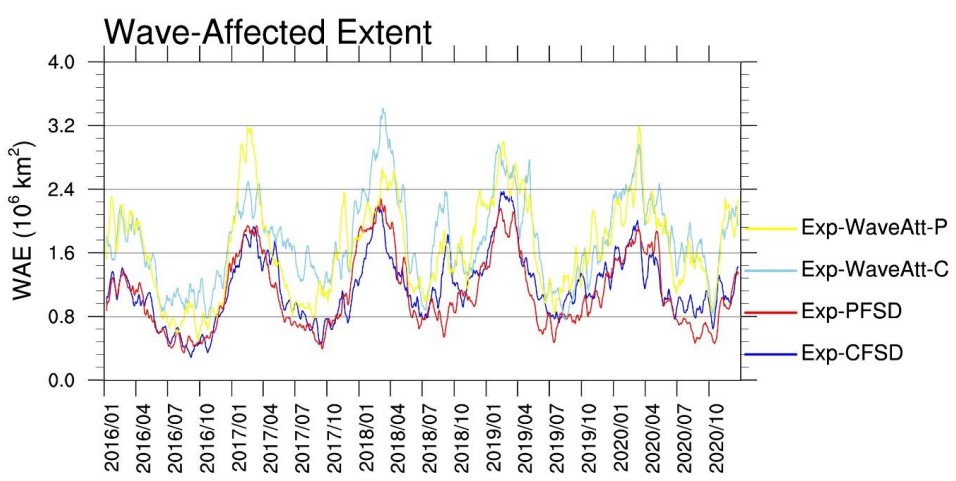

1121

Figure 11 Time-series (15-day running-averaged) of Arctic wave-affected extent for Exp-CFSD
(blue line), Exp-PFSD (red line), Exp-WaveAtt-C (light-blue line) and Exp-WaveAtt-P (yellow
line).