# Peer review of "Understanding influence of ocean waves on Arctic sea ice simulation: A modeling study with an atmosphere-ocean-wave-sea ice coupled model"

_The Cryosphere, 2023_

## Author Comment (AC1)

**Response to the reviews of tc-2023-79** "Understanding influence of ocean waves on Arctic sea ice simulation: A modeling study with an atmosphere-ocean-wave-sea ice coupled model" by Chao-Yuan Yang, Jiping Liu, Dake Chen

**Responses to comments by Reviewer #1**

**We would like to thank the reviewer for the helpful comments on the paper.**

With the reduction of Arctic sea ice, the expansion of summer sea ice marginal ice zone, and the enhanced moveability of sea ice, understanding influence of ocean waves on Arctic sea ice simulation and the role of floe size on the wave-ice interactions and ice-ocean heat exchange is becoming more important for both research communities from the Arctic sea ice numerical simulation and other related disciplines. This manuscript by Yang et al. investigates the impacts of ocean waves on Arctic sea ice simulation based on a newly-developed atmosphere-ocean-wave-sea ice coupled model. Especially, the contrasting behaviors of floe size, the responses of sea ice to different lateral melting rate formulations, and the sensitivity of sea ice to the simulated wave parameters have been investigated in detail. This is a work worth publishing. However, there are still some confusions that need further revision and clarification. Therefore, I recommend that this paper be considered for publication after minor revisions.

The major point of concern is the simulation effect of oceanic mixed layer. This paper discusses the reshaping process of sea waves on the size of floating ice, as well as the impact of the latter on ice-ocean heat exchange. Then I think the simulation effect of the ocean mixed layer must be discussed, so I suggest adding 1-2 illustrations to compare the simulation results of the depth and heat content of the mixed layer under different mode settings, and discuss on their influence on ice-ocean heat flux.

**Response: Thanks for the reviewer's helpful comment. In this revision, we determined**

the mixed layer depth (MLD) based on 0.1 degree Celsius difference relative to the surface temperature (e.g., Courtois et al., 2017, their Table 2). The choice of the temperature difference method for MLD determination is due to the calculation of heat content within MLD, which is mainly controlled by the temperature difference between ocean temperature and freezing point. Figure R1-R4 show the monthly-mean of MLD in March and September for all experiments conducted in this study with the same grouping described in Section 3 of the manuscript. In general, as shown in Fig. R1-R4, all experiments exhibit similar evolution of MLD, that is MLD is deeper (up to 150m) in March and shallower (up to 80m) in September. MLD in the open waters is broadly similar across all experiments and MLD near the ice edge (15% ice concentration, black contour in Fig. R1-R4) is shallower (10-30m) relative to other areas. In March, MLDs under ice-covered areas become deeper as lead time increases.

To calculate the heat content within MLD, we used the same approach for the calculation of melting potential in the ROMS model as described in Smith et al. (2010), which is defined as the vertical integral from surface to MLD of the difference between ocean temperature and freezing point. The calculated values of heat content/melting potential have the same unit ($W/m^2$) and directionality (positive downward) as ice-ocean heat flux, and they represent the "maximum" heat flux that the ice can extract. Figure R5 and R6 show the heat content of MLD and melting potential for Exp-CFSD and Exp-PFSD in March and September. As shown in Fig. R5-R6, Exp-PFSD shows less the melting potential (0-5m) and the heat content within MLD under ice-covered areas compared to Exp-CFSD. This feature is more pronounced in September than in March. Also, heat content in MLD near ice edge of Exp-PFSD reduces more than other ice-covered areas compared to that of Exp-CFSD, suggesting the role of ice-ocean heat flux. Figure R5 and R6 further support the constraint role of limited oceanic energy to ice melting with respect to varied floe-size not only in the surface layer (i.e., melting potential) but also in

**the mixed layer. The above analyses and discussions were added in the revised manuscript L625-L646.**

*Reference:*

*Courtois, P., Hu, X., Pennelly, C., Spence, P., and Myers, P. G.: Mixed layer depth calculation in deep convection regions in ocean numerical models, Ocean Modelling, 120, 67-78, http://dx.doi.org/10.1016/j.ocemod.2017.10.007, 2017.*

*Smith, R., Jones, P., Briegleb, B. P., Bryan, F. O., Danabasoglu, G., Dennis, J. M., Dukowicz, J., Eden, C., Fox-Kemper, B., Gent, P., Hecht, M., Jayne, S., Jochum, M., Large, W., Lindsay, K., Maltrud, M., Norton, N., Peacock, S., Vertenstein, M., and Yeager, S.: The Parallel Ocean Program (POP) reference manual: Ocean component of the Community Climate System Model (CCSM), https://opensky.ucar.edu/islandora/object/manuscripts%3A825/, 2010.*

[Figure]

**Figure R1 Monthly-mean of MLD in March (top panel) and September (bottom panel) of Exp-CFSD and Exp-PFSD for 2016-2020. Note: black contour represents the averaged location of 15% ice concentration.**

[Figure]

**Figure R2 Same as Figure R1 but for Exp-CFSD, Exp-PFSD, Exp-LatMelt-C, and Exp-LatMelt-P.**

[Figure]

**Figure R3 Same as Figure R1 but for Exp-CFSD, Exp-PFSD, and Exp-WaveFrac-P.**

[Figure]

**Figure R4 Same as Figure R1 but for Exp-CFSD, Exp-PFSD, Exp-WaveAtt-C, and Exp-WaveAtt-P.**

**Figure R5 March-averaged heat content of MLD (top panel) and melting potential (bottom panel) of Exp-CFSD and Exp-PFSD for 2016-2020. Note: black contour represents the averaged location of 15% ice concentration.**

[Figure]

**Figure R6 September-averaged heat content of MLD (top panel) and melting potential (bottom panel) of Exp-CFSD and Exp-PFSD for 2016-2020. Note: black contour represents the averaged location of 15% ice concentration.**

Other special comments:

Line 56 "the ice-floe melting rate is a result of the interaction between floe size and ocean circulation": Why the floe size interacts with the ocean circulation is not clear here, what scale of ocean circulation is, and why it affects the ice-ocean heat flux and ice melt rate?

**Response: Horvat et al. (2016) set up idealized experiments based on MITgcm with different total numbers of ice floe with the same size uniformly spaced in a 75*75 km**

domain, these experiments have the same initial total ice coverage (49.92% of the domain) and ice volume, and ice momentum equation is disabled. At the early stage of simulation, relatively cold and fresh water is formed in the ice-covered areas and leads to density gradient between ice-covered and ice-free regions, and then geostrophic surface currents along the floe edge. Within several days of simulation, baroclinic instabilities appear along the density gradient, mix ocean energy from ice-free areas (where have net incoming heat flux from the atmosphere by design) to under-floe areas, and finally lead to enhanced ice melting. The enhancement of ice melting rate is increasing with the number of floes (i.e., as floe size decreases). Different from Horvat et al. (2016), who only investigated thermodynamics melt of ice floes, Gupta and Thompson (2022) further consider both mechanical and thermodynamics effects on ice floes and also shows the ice melting rate is related to floe size. In this revision, we modified the text to better reflect the processes described above and now it reads as "Some studies also show that the ice-floe melting rate is associated with horizontal mixing of oceanic heat across ice floe edge between open water and under-floe ocean by oceanic eddies, in particular sub-mesoscale eddies, and the strength of this effect depends on floe size (Gupta and Thompson, 2022; Horvat et al., 2016)." in the revised manuscript L55-L58.

Line 62 "Previous studies showed that intense storms like "Great Arctic Cyclone" of 2012 (Simmonds and Rudeva, 2012) and strong summer cyclone in 2016 contribute to the anomalously low sea ice extent in 2012 and 2016": This can only be said to be a partial contribution, as even without great cyclones, there will be an extremely low Arctic sea ice extent in the summer of 2012.

**Response: Thanks for the reviewer's comment. We changed this sentence to "Previous studies suggested that intense storms like "Great Arctic Cyclone" of 2012 (Simmonds and**

**Rudeva, 2012) and strong summer cyclone in 2016 could be one of contributors to the anomalously low sea ice extent in 2012 and 2016…" in the revised manuscript L64-L66.**

Line 96 "a full representation of sea ice responses under the interactions across atmosphere, ocean, wave, and sea ice": Actually, waves are a part of the ocean.

**Response: Thanks for the reviewer's suggestion. We modified the text to "… a full representation of sea ice responses to the evolving states of atmosphere, ocean, and wave based on explicit model physics as well as feedbacks from sea ice to them" in the revised manuscript L99-L101.**

frazil ice formation: the frazil ice is very discrete, how does it affect the air-ocean heat flux?

**Response: We follow the nomenclature in the documentation of Sea Ice Model Intercomparison Project (Notz et al., 2016, Append. E), which defines sea ice mass change through ice growth in supercooled open water (a.k.a. frazil ice formation) as one of the sea ice mass budget terms. In this study, frazil ice formation is equivalent to any newly-formed ice mass by supercooled water in the CICE model.**

*Reference:*

*Notz, D., Jahn, A., Holland, M., Hunke, E., Massonnet, F., Stroeve, J., Tremblay, B., and Vancoppenolle, M.: The CMIP6 Sea-Ice Model Intercomparison Project (SIMIP): understanding sea ice through climate-model simulations, Geosci. Model Dev., 9, 3427–3446, https://doi.org/10.5194/gmd-9-3427-2016, 2016.*

The floe welding parameter: How to consider the seasonal changes in the welding coefficient

of sea ice, especially during the freeze-thaw transition season, whether it will be affected by ice temperature and thickness? If the seasonal variation of welding coefficient is considered, how will it affect the simulation results ?

**Response: As described in Roach et al. (2018), the floe welding process only occurs in the freezing condition. In this study, the freezing condition of each cell is determined by the net ice mass increase based on sea ice mass budget terms excluding the dynamics term. Then, the floe welding parameter acts like a step function once the freezing condition is met, and its value changes from 0 to the prescribed value (section 3 in the manuscript). In addition to the step function-like behavior of the floe welding parameter, sea ice concentration also contributes to seasonal changes of the floe welding process from the formulation perspective. The floe welding process is parameterized as (Roach et al., 2018),**

$$\frac{\partial N}{\partial t} = -\frac{\kappa}{2}C^2$$

**where $N$ is floe number density, $\kappa$ is the floe welding parameter, and $C$ is sea ice concentration, and the changes in $N$ is proportional to the square of $C$. Combined the step-change of the floe welding parameter between the freeze-thaw transition and the seasonal signal of ice concentration, the seasonal changes in the floe welding process are considered in this study. However, whether the floe welding parameter itself is a function of other variables (e.g., ice thickness, ice temperature) is still an open question due to limited field observations and laboratory experiments. The duration of contact between floes, the heat loss from the floes, or the overlap area between floes might be also important for the floe welding process (e.g., Manucharyan and Montemuro, 2022; Shen and Ackley, 1991). In this revision, we added more descriptions for the freeze-thaw transition of the welding parameter in Section 3 of the revised manuscript L278-L282.**

*Reference:*

*Manucharyan, G. E., and Montemuro, B. P.: SubZero: A sea ice model with an explicit representation of the floe life cycle. Journal of Advances in Modeling Earth Systems, 14, e2022MS003247. https://doi.org/10.1029/2022MS003247, 2022.*

*Roach, L. A., Smith, M. M., and Dean, S. M.: Quantifying growth of pancake sea ice floes using images from drifting buoys. Journal of Geophysical Research: Oceans, 123, 2851–2866. https://doi.org/10.1002/2017JC013693, 2018.*

*Shen, H. H., and Ackley, S. F.: A one-dimensional model for wave-induced ice-floe collisions. Annals of Glaciology, 15, 87–95. https://doi.org/10.1017/s0260305500009587, 1991.*

**Response to comments by Reviewer #2**

**We would like to thank the reviewer for the helpful comments on the paper.**

This study quantifies the effect of ocean waves on sea ice simulation in the arctic based on a coupled model framework built by the authors. Authors focus on the floe size and thickness distribution (FSTD) with the effect of the ocean waves embedded. This study demonstrates that involving wave-related process can have an impact on sea ice, proving the importance of oceanic wave on sea ice modeling in the coupled model.

Overall, the model development work in this study has a significant value on the coupled modeling system. The result in this study offers more insights on the interaction between the atmosphere, ocean, and sea ice in the arctic. The whole manuscript is well-written in general, and I recommend an acceptance after some minor revisions.

Major points:

In this study, the authors divide the model domain into 3 sub-regions, while it lacks conclusions

that related to geographically-specified features. I understand that the model development in this study has a good application for all these three regions, and it can distinguish the different wave-sea ice interactions in these regions. But authors should elaborate more on how the regional features derives the conclusion that are widely-applicable for the pan-arctic.

**Response:**

**Thanks for the reviewer's constructive comment. In this revision, we added additional text and an additional figure to better illustrate geographically-specified features of 3 sub-regions. In the revised manuscript L435-L447, now it reads "… the strengthened vertical mixing brings warmer water of the subsurface upward and maintains/increases the melting potential in the subregions. Figure 8d-f also show that the warmer signal in the upper ocean (at least to 60m depth) of Exp-PFSD persists after July, 2018 in the ATL region while the LK and BCE regions show seasonal oscillation of ocean temperature in the upper ocean for the entire simulation. Combined with the regional SIA shown in Figure 7d-f, seasonal fully ice-covered states in the LK and BCE regions force the upper ocean to restore to certain states (i.e., near freezing point under sea ice, near zero melting potential shown in Fig. 7k-l) for both Exp-CFSD, and Exp-PFSD, which might mitigate the effects of ocean wave activities and other processes on the upper ocean. Figure 9(R7) shows the difference of dynamical and thermal mass change between Exp-PFSD and Exp-CFSD. With less restoring effect by sea ice on the upper ocean in the ATL regions, the difference of thermally-induced mass change between Exp-PFSD and Exp-CFSD shows a larger variation once the upper ocean difference starts to persist after July, 2018 (Fig. 8d, 9d) while the variations in the LK and BCE regions remain relatively unchanged for the entire simulation (Fig. 9e-f)."**

[Figure]

**Figure R7 Time-series of the difference of (a-c) dynamical mass change and (d-f) thermal mass change between Exp-PFSD and Exp-CFSD in the ATL, LK, and BCE regions.**

Minor points

Line 238: Please specify that if the atmosphere, ocean, and sea ice model are using the same model grid.

**Response: Thanks for the reviewer's comment. We changed the sentence to "The WRF, ROMS, SWAN, and CICE models use the same model grid with 320 (440) x- (y-) grid points and ~24km horizontal resolution (Fig. 1)" in the revised manuscript L242-L243.**

Line 248: The model configuration of a higher model top is kind of confusing to me. Does it only matter on the atmospheric circulation modeling? Or it has some effect on the better coupling between the atmosphere and ocean/sea ice?

**Response: Cassano et al. (2011) showed that a higher model top can reduce the bias in the simulated sea level pressure (SLP) based on the standalone WRF model. Without an elevated model top, and the circulation biases exhibit not only in SLP but also enhance with height. They suggested that this top-down bias in the circulation is associated with**

the model-top boundary treatment, which is also shown in other modeling studies (ARCSyM, Lynch and Cullather, 2000; HadAM3, Scaife et al., 2005). In our preliminary multiyear simulations with our coupled model before conducting this study, the higher model top can lead to better simulated ice mass distribution, which might be able to interpreted as better coupling between the atmosphere and sea-ice.

*Reference:*

*Cassano, J. J., Higgins, M. E., and Seefeldt, M. W.: Performance of the Weather Research and Forecasting Model for Month-Long Pan-Arctic Simulations. Monthly Weather Review, 139, 11, 3469-3488, https://doi.org/10.1175/MWR-D-10-05065.1, 2011.*

*Lynch, A. H., and Cullather, R. I.: Investigation of boundary forcing sensitivities in a regional climate model. J. Geophys. Res., 105, 26603–26617, 2000.*

*Scaife, A. A., Knight, J. R., Vallis, G. K., and Folland, C. F.: A stratospheric influence on the winter NAO and North Atlantic surface climate. Geophys. Res. Lett., 32, L18715, doi:10.1029/2005GL023226, 2005.*

Figure 9 make sure the naming of sub-figures correctly follows the rule of TC.

**Response: Thanks for the reviewer's suggestion. We re-plotted all figures using sub-indexing for the naming in this revision as shown in Figure R8-R10.**

**Sea Ice Concentration (1)**

(a) Exp-CFSD 202009  (b) Exp-PFSD 202009  (c) Exp-LatMelt-C 202009  (d) Exp-LatMelt-P 202009

5 10 15 20 25 30 35 40 45 50 55 60 65 70 75 80 85 90 95

**Sea Surface Temperature (°C)**

(e) Exp-CFSD 202009  (f) Exp-PFSD 202009  (g) Exp-LatMelt-C 202009  (h) Exp-LatMelt-P 202009

[Figure]

-0.9 -0.6 -0.3 0 0.3 0.6 0.9 1.2 1.5 1.8 2.1 2.4 2.7 3

**Friction Velocity (10⁻³ m/s)**

(i) Exp-CFSD 202009  (j) Exp-PFSD 202009  (k) Exp-LatMelt-C 202009  (l) Exp-LatMelt-P 202009

[Figure]

0.5 0.6 0.7 0.8 0.9 1 1.1 1.2 1.3 1.4 1.5

**Figure R8 The monthly-mean of (a-d) sea ice concentration, (e-h) sea surface temperature, and (i-l) friction velocity in September, 2020 for Exp-CFSD, Exp-PFSD, Exp-LatMelt-C, and Exp-LatMelt-P.**

[Figure]

**Figure R9** Time-series of (a-c) ice extent, (d-f) ice area, (g-i) significant wave height, (j-l) melting potential, and (m-o) heat flux at the ocean surface in ATL, LK, and BCE regions for Exp-CFSD (blue line) and Exp-PFSD (red line). Note: significant wave height, melting potential, and heat flux at the ocean surface are region-averaged and 15-day running-averaged values.

[Figure]

**Figure R10 The spatial distribution of the representative floe radius in March (upper panel) and September (bottom panel) of (a-e) Exp-PFSD, (f-j) Exp-LatMelt-P, (k-o) Exp-WaveFrac-P, and (p-t) Exp-WaveAtt-P for 2016-2020. Note: cells with less than 15% ice concentration are treated as missing values.**

---

## Author Response (AR2)

**Response to the reviews of tc-2023-79** "Understanding influence of ocean waves on Arctic sea ice simulation: A modeling study with an atmosphere-ocean-wave-sea ice coupled model" by Chao-Yuan Yang, Jiping Liu, Dake Chen

**Responses to comments by Reviewer #1**

**We would like to thank the reviewer for the helpful comments on the paper.**

Review on "Understanding influence of ocean waves on Arctic sea ice simulation: A modeling study with an atmosphere-ocean-wave-sea ice coupled model" by Yang et al.,

The authors have made detailed revisions according to the previous-round review comments, and the manuscript has basically reached the state of being ready for publication. Here, I mainly raise some specific comments on unclear expressions and editorial issues:

1) The spelling rules for the author's name in English are not consistent for all authors. I understand that Chaoyue is the the first name as Jiping and Dake.

**Response: The spelling for the authors' first name in English is consistent with the spelling on their passport.**

2) Line 24 and other text: Sea-ice: I don't think it's necessary to use hyphens.

**Response: Thanks, we changed them.**

3) Line 54: The increased surface wind speed: So far, there is no study indicating an increasing trend in the near surface wind field over the Arctic Ocean (e.g., Zhang et al., 2022 as you cited)

**Response: Thanks for the reviewer's helpful comment. We mainly want to address the correlation between wave height, fetch and surface wind as shown in the previous studies. We changed the sentences to "Previous studies suggested that the Arctic fetch and surface wind speed over the ice-free ocean correlate well with wave heights in the Arctic Ocean (e.g., Casas-Prat and Wang, 2020; Dobrynin et al., 2012; Liu et al., 2016; Stopa et al., 2016; Waseda et al., 2018)."**

4) Line 58: Sea ice with mostly smaller floes has larger surface areas, particularly lateral surfaces.-- This expression makes me very vague. It would be better change to: The ice pack, with the same concentration, has larger surface area for the ice floes with smaller sizes, particularly lateral surfaces.

**Response: Thanks for the suggestion. We changed the sentence.**

5) Line 97-99: The coupled effects of ocean waves and sea ice include; the amplitude of ocean waves decays as the waves travel under the ice cover due to the combination of scattering and dissipation-- These two sentences appear incomplete and incoherent. It would be better change to: The coupled effects of ocean waves and sea ice also include the decay of amplitude of ocean waves as they travel under the ice cover due to the combination of scattering and dissipation.

**Response: Thanks. We changed the sentences.**

6) Line 337-342: In the equation (2), the user-defined coefficients for the wave attenuation are set as $c_2 = 1.06 \times 10^{-3}$ and $c_4 = 2.3 \times 10^{-2}$ (case 1), which follow the polynomial of Meylan et al. (2014, hereafter M14) from the observations with 10-25m floe in diameter in the Antarctic, and $c_2 = 2.84 \times 10^{-4}$ and $c_4 = 1.53 \times 10^{-2}$ (case 2), which follow the polynomial of Rogers et al. (2018, hereafter R18) based on the observations for pancake and frazil ice in the Arctic. --These parameters are all derived from the areas close to the ice edge. How applicable are they to the regions with higher ice concentration?

**Response: In this study, these user-defined wave attenuation parameters are uniform for the entire domain, and their effects are scaled by sea ice concentration of each cell (Equ. 3). Also, the choice of different parameters is mainly trying to qualitatively investigate how sea ice responds to different wave strength under sea ice. As we addressed in the**

discussion section, the attenuation by sea ice is associated with ice properties. A recent observational study also suggested that ocean waves decay faster under ice thickness over 0.5 meter (Huang and Li, 2023). That is, the parameters applied in this study may not be applicable to the regions with thicker ice.

*Huang, B. Q., and Li, X.-M.: Wave attenuation by sea ice in the Arctic marginal ice zone observed by spaceborne SAR. Geophysical Research Letters, 50, e2023GL105059. https://doi.org/10.1029/2023GL105059*

7) Line 410 the larger total ice surface area due to smaller floe size increases the efficiency of Exp-PFSD extracting energy from the ocean-- This process should generate positive feedback, That is to say, the lateral melting of sea ice would reduce the sea ice area, and then increase the absorption of heat (shortwave radiation) by the upper ocean, promoting sea ice melting, including at the ice bottom. Therefore, my question here is whether the numerical model can capture this feedback, that is, although some of the heat content is consumed by sea ice melting, it will also promote the ocean to absorb heat from the atmospheric in the warm melting season.

**Response: The average of all ice-cells (e.g., Fig. 3, 4, 5) may not be clear to see this increased energy input from the atmosphere to the ocean in the warm melting season that compensates the energy consumption by the increased ice surface area. By changing the view based on the regional average (e.g., Fig. 7), it is clear that less sea ice area can lead to more energy absorbed by the ocean in the melting season. That is, the model can capture this feedback in certain degree.**

8) Line 414 "Energy loss from the ocean through air-sea heat flux that further cools the upper ocean"-- As with the previous question, in summer, the increase in ocean area does not necessarily mean the loss of heat.

**Response: Thanks for the reviewer's comment. We modified the sentence and now it reads "Energy loss from the ocean through air-sea heat flux in winters that further cools the upper ocean, freshwater input (e.g., ice melting, precipitation) that raises the freezing point, as well as non-physical numerical oscillations (Naughten et al., 2017; Yang et al., 2022), are potential contributors that lead to increased frazil ice formation of Exp-PFSD as shown in Fig. 3a-b and Fig. S2g."**

9) Line 478 "), and more melting potential in December 2017 in the LK region": I have a question here. In December, shouldn't the LK region be covered fully by sea ice? Will sea ice melting occur in the early winter season here?

**Response: In Figure 7b, e, the LK region in Exp-PFSD is still not fully-covered by sea ice in December. In the preceding summer, less sea ice area in the LK region results in more energy stored in the ocean (Fig. 7k), which likely delays the ice growing in the LK region in December and leads to the corresponded sea ice area and melting potential shown in Fig. 7b, k.**

10) Line 502 "Combined with the warmer upper ocean in Exp-PFSD after January 2020 in the ATL region" ——In spring 2020, the sea ice transported from the Arctic Ocean to the Greenland and Barents seas should have increased due to abnormal atmospheric circulation (Zhang et al., 2023). Can the impact of the increased sea ice outflow from the central Arctic Ocean on the local mass budget of sea ice and heat fluxes be captured in numerical models?

Zhang, F., et al., 2023: The impacts of anomalies in atmospheric circulations on Arctic sea ice outflow and sea ice conditions in the Barents and Greenland seas: case study in 2020, The Cryosphere, 17, 4609–4628, https://doi.org/10.5194/tc-17-4609-2023.

**Response: The fully-coupled configuration, model biases, and limited constraints from boundary conditions of the WRF, ROMS models make the simulations analyzed in this**

study may not capture the actual changes in the Arctic during the period of 2016-2020. Further efforts are required to improve the fully-coupled model as we addressed in the discussion section so that the model can be better reproduce the changes in the Arctic.

11) Figure 1: Is the terminology of the subregions appropriate as it includes not only the peripheral seas but also the central Arctic Ocean, extending to 85N? For example, the LK region includes most of the basin areas located north of the LK Seas, and the same applies to other subregions.

**Response: We agree with the reviewer's comment that the spatial coverage extended to 85N also includes the part of the central Arctic Ocean. We also have used different coverage (e.g., 80N) and the behaviors shown in Fig. 7 and Fig. 8 remain similar to current spatial coverage. We modified the description for the subregions and now it reads "Based on geographic features, we define the following subregions for further analysis: 1) Barents and Greenland Seas (ATL, 45W-60E, 65N-85N), 2) Laptev and Kara Seas (LK, 60E-150E, 65N-85N), and 3) Beaufort, Chukchi, and East Siberian Seas (BCE, 150E-120W, 65N-85N, see black boxes in Fig. 1 for the geographic coverage of subregions). …The ATL region is only partially-limited by ice-covered areas while the LK and BCE regions can be fully-covered by sea ice in winter. Though these subregions also include part of the central Arctic Ocean, they will still be addressed by the main peripheral seas in the subregions in the following discussion for simplicity."**